# Optimized libraries for CRISPR-Cas9 genetic screens with multiple modalities

Kendall R. Sanson[1], Ruth E. Hanna[1], Mudra Hegde [1], Katherine F. Donovan[1], Christine Strand [1], Meagan E. Sullender [1], Emma W. Vaimberg[1], Amy Goodale[1], David E. Root[1], Federica Piccioni [1] & John G. Doench [1]

The creation of genome-wide libraries for CRISPR knockout (CRISPRko), interference (CRISPRi), and activation (CRISPRa) has enabled the systematic interrogation of gene function. Here, we show that our recently-described CRISPRko library (Brunello) is more effective than previously published libraries at distinguishing essential and non-essential genes, providing approximately the same perturbation-level performance improvement over GeCKO libraries as GeCKO provided over RNAi. Additionally, we present genome-wide libraries for CRISPRi (Dolcetto) and CRISPRa (Calabrese), and show in negative selection screens that Dolcetto, with fewer sgRNAs per gene, outperforms existing CRISPRi libraries and achieves comparable performance to CRISPRko in detecting essential genes. We also perform positive selection CRISPRa screens and demonstrate that Calabrese outperforms the SAM approach at identifying vemurafenib resistance genes. We further compare CRISPRa to genome-scale libraries of open reading frames (ORFs). Together, these libraries represent a suite of genome-wide tools to efficiently interrogate gene function with multiple modalities.

[1] Broad Institute of Harvard and MIT, 75 Ames Street, Cambridge, MA 02142, USA. These authors contributed equally: Kendall R. Sanson, Ruth E. Hanna Correspondence and requests for materials should be addressed to J.G.D. (email: jdoench@broadinstitute.org)

With robust on-target activity and high fidelity, CRISPR-Cas9 technology has surpassed RNAi as the preferred method for genetic screening[1,2]. Unlike RNAi, CRISPR technology enables screens beyond gene down-regulation: unmodified Cas9 can generate complete loss-of-function alleles, and nuclease-deactivated Cas9 (dCas9) may be tethered to inhibitory or activating domains to regulate gene expression via CRISPR interference (CRISPRi) or CRISPR activation (CRISPRa), respectively[3–8]. These technologies, in contrast to CRISPR knockout (CRISPRko), allow for the transient modulation of gene expression, which can reveal novel phenotypes and enable more flexible experimental designs.

Although early CRISPR libraries outperformed RNAi libraries, the selection of highly active sgRNAs can further improve library performance[9–11]. Numerous approaches have been developed to predict sgRNA activity[12] and deployed to create optimized genome-wide CRISPRko libraries for pooled screening[13–17], as well as genome-wide libraries for CRISPRi[18,19] and CRISPRa[18–20]. Although some libraries contain many sgRNAs per gene, allowing high confidence detection of hits, screening efficiency can also be improved by considering libraries holistically; well-designed libraries that effectively modulate targeted genes with fewer sgRNAs per gene provide more information with fewer resources. This is particularly useful in settings where cell numbers are limiting, such as screens in primary cells or in vivo.

Here, we evaluate genome-wide libraries for CRISPRko, CRISPRi, and CRISPRa with S. pyogenes Cas9 (SpCas9) via 14 screens across 3 cell lines. We present the dAUC (delta area under the curve) metric, a size-unbiased metric for library performance in negative selection, loss-of-function screens, which quantifies the ability of a genome-wide library to distinguish essential and non-essential genes. In negative selection screens, we demonstrate that Brunello, our optimized CRISPRko library, outperforms previously published CRISPRko libraries at both the sgRNA and the gene level. We also introduce two new human genome-wide libraries: Dolcetto (CRISPRi) and Calabrese (CRISPRa). We demonstrate that Dolcetto outperforms existing CRISPRi libraries in negative selection screens and performs comparably to CRISPRko in depletion of essential genes. In positive selection screens, we show that Calabrese finds more vemurafenib resistance genes than the SAM library approach[20]. Additionally, we compare CRISPRa to screens performed with open reading frame (ORF) overexpression libraries and show that these technologies identify both common genes and distinct hits. Our findings demonstrate the importance of optimized library design to improving the quality of genetic screens with CRISPRko, CRISPRi, and CRISPRa.

## Results

**Optimized genome-wide CRISPRko library**. We previously reported the design of optimized CRISPRko sgRNA libraries with improved on-target and reduced off-target activity in the human and mouse genomes, named Brunello and Brie, respectively[16]. The Brunello library comprises 77,441 sgRNAs, an average of 4 sgRNAs per gene, and 1000 non-targeting control sgRNAs. We conducted genome-wide negative selection (dropout) screens in A375 melanoma cells that were first engineered to express Cas9. The Brunello library in the lentiGuide vector[2] was transduced into cells in biological replicates at a multiplicity of infection (MOI) of ~0.5 and passaged at a minimum of 500x coverage; that is, the majority of transduced cells received only a single viral integrant and each sgRNA is present, on average, in 500 unique cells. Uninfected cells were removed with puromycin selection, and the population was cultured for a total of 3 weeks, after which

genomic DNA was harvested, the sgRNA cassette retrieved by PCR, and the abundance of sgRNAs quantitated by Illumina sequencing (Supplementary Data 1). All pooled screens described herein follow these general experimental parameters.

To evaluate performance, we calculated the area under the curve (AUC) for all sgRNAs targeting the gold-standard gene sets of 1580 essential and 927 non-essential genes[15,21] (Fig. 1a). An effective library should have an AUC>0.5 for sgRNAs targeting essential genes, indicating that these sgRNAs preferentially deplete relative to their starting abundance, and an AUC ≤ 0.5 for sgRNAs targeting non-essential genes and non-targeting sgRNAs, indicating that these preferentially remain. Importantly, because the AUC metric includes every sgRNA without prior filtering for the top-performing sgRNAs, it is not biased towards larger libraries and can discriminate the quality of sgRNA design across libraries of different sizes.

In A375 cells, which were screened previously with the GeCKO and Avana libraries[2,16], the Brunello library showed greater depletion of sgRNAs targeting essential genes (AUC = 0.80), while sgRNAs targeting non-essential genes showed no evidence of depletion (AUC = 0.42; Fig. 1b). Conversely, non-targeting sgRNAs were among the least depleted (AUC = 0.16), evidence of the well-described cutting effect in CRISPRko screens, whereby dsDNA breaks lead to detectable effects on cell growth; this is magnified in extreme cases such as copy number amplified target sites or promiscuous sgRNAs[16,22–24].

To simplify comparisons across libraries, we next calculated the difference between the AUC of sgRNAs targeting essential and non-essential genes (delta AUC, dAUC). The dAUC increased with each generation of CRISPRko library design, from GeCKO to Avana to Brunello, validating that improvements in sgRNA design led to increased performance; combining data from multiple replicates led to minor increases in dAUC, suggesting that individual replicates were well-powered (Fig. 1c). Notably, the improvement in dAUC from GeCKOv2 to Brunello (delta dAUC, ddAUC = 0.46–0.24 = 0.22) was greater than the average improvement from RNAi to GeCKOv2 in Project Achilles[25] (ddAUC = 0.17).

We then calculated dAUCs for previously published CRISPRko libraries (Supplementary Table 1)[10,14,15,17,23,24,26] and found that Brunello outperformed all others by this metric (Fig. 1c). The TKOv3 library, screened in the haploid cell line HAP1, was the next-best performer[10]. Importantly, the design rules underlying Brunello were not trained on data from negative selection screens[16], which may also select for promiscuous sgRNAs, and thus the dAUC metric represents an unbiased performance measurement for this library. We additionally compared dAUCs calculated with the initial essential gene set[21], which included data from RNAi screens only, to those calculated with the updated set[15], which also considered data from initial CRISPRko screens. We observed a strong correlation, indicating that the dAUC metric is not inflated by the use of an essential gene set informed by CRISPRko screens (Supplementary Figure 1a).

To compare libraries at the gene level, we averaged all sgRNAs targeting a gene and performed precision-recall analysis, defining essential genes as true positives and non-essential genes as false positives, and calculated the area-under-the-curve of the receiver-operator characteristic (ROC-AUC) (Fig. 1d). In contrast to the dAUC metric, the ROC-AUC metric highlights the value of having more sgRNAs per gene; for example, GeCKOv1 and GeCKOv2 used the same design criteria and thus have similar dAUC values, but GeCKOv2, with 6 sgRNAs per gene, substantially outperformed GeCKOv1, with 3–4 sgRNAs per gene, via the ROC-AUC metric (Fig. 1e, Supplementary Figure 1b). Notably, Brunello contains only 4 sgRNAs per gene but outperformed all other libraries by both metrics.

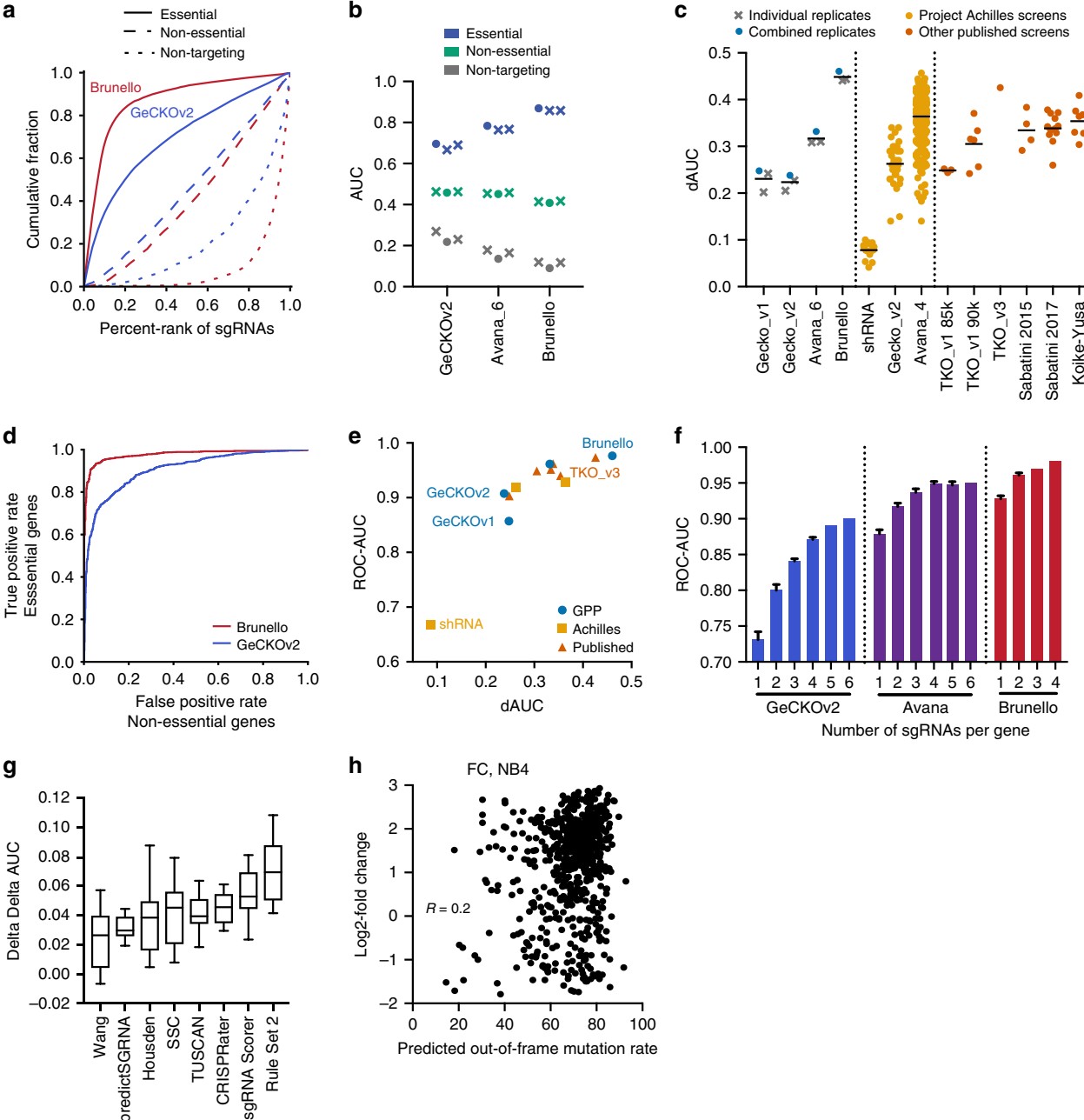

**Fig. 1** Improved performance of genome-wide CRISPRko libraries. **a** Area under the curve analysis of cell viability screens for core essential (solid line), non-essential (dashed line), and non-targeting (dotted line) gene sets in the Brunello and GeCKOv2 library screened in A375 cells. **b** Comparison of AUCs for essential, non-essential, and non-targeting sgRNAs across the generations of libraries screened by this group. The AUC of individual replicates are plotted as X's, while the AUC values calculated from the averaged log2-fold change of the replicates are plotted as circular points. The Avana library was screened with 6 sgRNAs per gene. **c** Comparison of the dAUC across different CRISPRko libraries. All published libraries are plotted as circular points from the combination of replicates, if provided. Black line represents the average of the dAUCs. For Project Achilles, a version of the Avana library with 4 sgRNAs per gene was used; the shRNA data are shown for the same 33 cell lines that were screened with GeCKOv2. **d** Receiver operating characteristic analysis of cell viability screening data for the Brunello and GeCKOv2 libraries screened in A375 cells. False positive rates are determined by non-essential genes and plotted against the true positive rate, determined by essential genes. **e** Comparison of the dAUC and ROC-AUC values for different libraries; when multiple cell lines of data were available, the mean is plotted. GPP refers to screens described here and previously by this group. **f** Subsampling analysis calculating the ROC-AUC of n sgRNAs per gene in three different libraries. Error bars represent s.d. calculated from different iterations of library sampling. **g** Difference in the dAUC (ddAUC) for various scoring schemes when applied to the 33 cell lines screened with GeCKOv2. The box represents the 25th, 50th, and 75th percentiles; whiskers show 10th and 90th percentiles. **h** Comparison of the predicted out-of-frame mutation rate to the previously measured log2-fold change of sgRNAs in the flow cytometry (FC) dataset targeting cell surface genes in the NB4 cell line

We next performed subsampling analysis of sgRNAs in the Brunello, Avana, and GeCKOv2 libraries. After random draws of *n* sgRNAs per gene, sgRNAs targeting the same gene were averaged, and the ROC-AUC was calculated (Fig. 1f). Even with one sgRNA per gene, the Brunello library outperformed the GeCKOv2 library with six sgRNAs per gene, highlighting the effect of improved sgRNA design. We observed only a minor increase in performance with additional sgRNAs for the Brunello library, suggesting that a library with only 2 or 3 highly effective sgRNAs per gene may still perform well.

To investigate whether factors such as technical execution could bias comparisons across libraries, we applied other published scoring schemes to the 33 cell lines in the Project Achilles dataset that were screened with GeCKOv2[23]. As this library was designed before the development of any sgRNA design rules, it provides a neutral dataset for testing scoring schemes. We selected the top 10% of sgRNAs defined by each scoring scheme, re-calculated the dAUC for these filtered subsets, and determined the difference from the dAUC of the entire library (ddAUC). We observed that filtering by Rule Set 2 provided the greatest increase in performance across these 33 cell lines (Fig. 1g, Supplementary Figure 1c)[13,16,27–32]. Notably, some scoring schemes were assembled using data from negative selection screens and thus this analysis may overestimate their performance, but Rule Set 2 was developed with no input from any viability screens[16]. We conducted a complementary analysis by applying our sgRNA selection scheme to published libraries and re-calcalculating the dAUC for only those sgRNAs ranked in the top 10% by our scoring criteria (Supplementary Figure 1d). For all libraries, the dAUC improved with the application of selection criteria used in the design of the Brunello library. These analyses indicate that the increased performance of the Brunello library is largely due to sgRNA selection rather than specific technical parameters of the screens presented here.

Finally, to test whether sgRNAs predicted to introduce out-of-frame (OOF) mutations show higher activity, we used FORECasT[33] to calculate the OOF mutation rate for sgRNAs in the previously published flow cytometry (FC) and small molecule resistance (RES) tiling datasets[16]. Overall, sgRNAs with a high OOF mutation rate exhibited a range of activity, suggesting that OOF mutation rate on its own is not highly predictive of activity (Fig. 1h; Supplementary Figure 2). However, this metric should be considered as a feature when developing future sgRNA design rules.

**Modifications to the tracrRNA.** While sgRNA design is critical to maximizing on-target activity, other components of CRISPR technology have also been optimized. Previously, several modifications to the tracrRNA have been proposed to increase on-target activity, including the removal of a potential RNA polymerase III termination site and extension of the tetraloop to enhance sgRNA-Cas9 complex assembly[34]. Additional studies demonstrated that a T-to-C and compensatory A-to-G substitution, along with a five nucleotide extension of the tetraloop, was optimal[35]. These modifications were suggested to improve the on-target activity of sgRNAs in small-scale tests[14] and in a CRISPRko screen with a focused library[36]. To date, however, there has been no genome-wide, head-to-head comparison of the modified tracrRNA, nor a thorough characterization of its effect on off-target activity.

We designed a modified tracrRNA for use in lentiCRISPRv2, hereafter called tracr-v2, which removed the Pol III termination site and extended the tetraloop by 5 base pairs (Fig. 2a). To test on- and off-target activity with tracr-v2, we designed a tiling library containing all possible sgRNAs targeting *EEF2*, a core

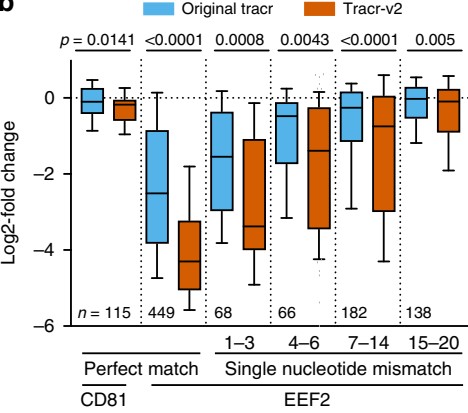

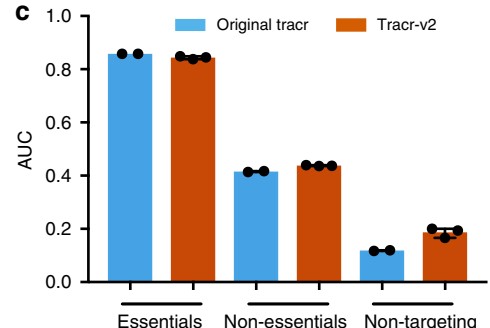

**Fig. 2** Evaluation of an alternative tracrRNA. **a** Comparison of the sequence of the original tracrRNA and tracr-v2. **b** Comparison of the log2-fold-change of perfectly matched sgRNAs and single base mismatches targeting the essential gene *EEF2* for the original tracrRNA and tracr-v2. *CD81*-targeting sgRNAs serve as the control. Mismatched sgRNAs are binned by position, and are numbered such that nt 20 is PAM-proximal. The box represents the 25th, 50th, and 75th percentiles, and whiskers show 10th and 90th percentiles. Two-tailed Welch's *t*-test was used to compare the distributions within each bin and *p*-values are indicated.The number of sgRNAs in each comparison is shown at the bottom. **c** AUC values for essential, non-essential, and non-targeting gene sets with the original tracrRNA and tracr-v2 in genome-wide viability screens. Error bars represent the range of two or three biological replicates for screens with the original and modified tracrRNA, respectively

essential gene, with both perfect match and single-mismatched sgRNAs; notably, this latter class was not previously assessed[36]. We performed negative selection screens in A375 cells with either the original tracrRNA or tracr-v2, using sgRNAs targeting *CD81* serving as controls, as done previously[37] (Supplementary Data 2). Consistent with other reports, perfect match sgRNAs with tracr-v2 showed significantly stronger depletion of *EEF2* than the original tracrRNA, indicating improved on-target activity (Fig. 2b). However, tracr-v2 also exhibited higher levels of off-target activity with mismatched sgRNAs (Fig. 2b).

To test how this increase in both on- and off-target activity would affect performance of a genome-wide screen, we performed a negative selection screen using Brunello with tracr-v2 in lentiCRISPRv2 (Supplementary Data 1). We observed that Brunello with tracr-v2 performed similarly to our previous results with the original tracrRNA (Fig. 2c); tracr-v2 had a dAUC of 0.42, compared to 0.46 with the original tracrRNA. These results suggest that tracr-v2 neither improves nor harms library performance in CRISPRko genome-wide screens, at least in a

library with highly active sgRNAs; we cannot rule out that a modified tracrRNA structure would have more beneficial impact on a library with less on-target activity.

**Optimized genome-wide CRISPRi library**. A useful orthogonal approach to study loss-of-function phenotypes is CRISPR interference (CRISPRi). Here, point mutations engineered into Cas9 inactivate the nuclease domains, creating an RNA-guided DNA binding protein (dCas9), which can then be tethered to repressive domains such as KRAB to prevent efficient transcription[4,7,8,18]. Although CRISRko can target a large portion of a protein coding gene, CRISPRi is effective over a constrained region, near the transcription start site (TSS). Previous studies have shown that the FANTOM database[38], which relies on Cap Analysis of Gene Expression (CAGE) to identify the TSS, provides more accurate annotations for CRISPRi sgRNA selection than either NCBI (RefSeq) or Ensembl, which identify the TSS using alternative approaches[39]. Based on our analysis of existing data[18,40], and consistent with previous findings[19,39], we identified the window of +25 to +75 nts downstream of the TSS as optimal for CRISPRi sgRNAs (Fig. 3a).

To design the library, we first selected sgRNAs in this optimal window, and further ranked them by Rule Set 2, which is effective for CRISPRi sgRNAs[16], and the number of off-target sites. In order to fulfill a quota of 6 sgRNAs per gene, we successively relaxed these three criteria (Supplementary Table 2). The resulting library, named Dolcetto, was divided into Sets A and B, with the former containing the top three selected sgRNAs. This library was cloned into a modified version of lentiGuide (pXPR_050); we opted to use tracr-v2 both because the limited window of CRISPRi activity may mitigate the risk of off-target effects and because previous CRISPRi studies have used a modified tracrRNA[18,19].

To assess Dolcetto performance, we performed negative selection screens in triplicate in A375 and HT29 cell lines stably expressing KRAB-dCas9[41] (Fig. 3b; Supplementary Data 3). We applied the dAUC (Fig. 3c) and ROC-AUC (Fig. 3d) metrics described above to compare sgRNA- and gene-level performance, respectively. In both cell lines, Set A performed better than Set B, indicating that the top three sgRNAs selected by our heuristic were indeed more likely to be active than the next three sgRNAs. We applied the same metrics to previously-published CRISPRi screens with the hCRISPRi-v2 library[19] in K562 cells (Supplementary Table 3), and observed substantially better performance with Dolcetto (Fig. 3c,d). Subsampling analysis (combining Set A and Set B for Dolcetto) showed that Dolcetto, with 3 sgRNAs per gene, outperformed hCRISPRi-v2 with 10 sgRNAs per gene (Fig. 3e), suggesting that the heuristics used to design this library were highly effective.

To further investigate library differences, we assigned the sgRNAs in hCRISPRi-v2 to the various selection rounds in our heuristic. We observed that a small fraction of hCRISPRi-v2 sgRNAs (17%) were in the first selection round, in contrast to Dolcetto (75%, Fig. 3f). Indeed, the dAUC for this filtered subset of the hCRISPRi-v2 library was higher than the full library, with the dAUC improving from 0.29 to 0.36 (Fig. 3c). We conclude that the differences in performance between hCRISPRi-v2 and Dolcetto can mostly be attributed to improved CRISPRi sgRNA design rather than differences across cell lines or experimental execution.

Finally, we asked whether the confidence of TSS annotation corresponds to sgRNA activity. As a proxy for TSS confidence for a gene, we calculated the fraction of CAGE peak reads assigned to the p1 promoter relative to all other annotated promoters. We observed that essential genes with a higher fraction of p1 CAGE peak reads were more depleted (Supplementary Figure 3a, b), as were essential genes whose p1 CAGE peaks had more total reads,

another metric of TSS confidence (Supplementary Figure 3c, d). These results highlight the importance of TSS annotation for effective CRISPRi activity.

**Comparison of CRISPRko and CRISPRi**. The dAUC and ROC-AUC metrics showed that Brunello and Dolcetto provided similar discrimination between essential and non-essential genes. We next examined the data for signs of cutting-related toxicity, as has been previously been reported to be present with CRISPRko[22–24] but not with CRISPRi[19,41]. For this analysis, we compared the AUCs for sgRNAs targeting non-essential genes, which should cut the genome with Brunello but not Dolcetto, with non-targeting sgRNAs, which should not cut the genome with either library (Fig. 4a). Brunello exhibited a distinct cutting effect, as demonstrated by a lower AUC for non-targeting sgRNAs (AUC = 0.09) compared to sgRNAs targeting non-essential genes (0.41), as has been detailed previously[10]. Dolcetto, however, showed little difference between non-essential (AUC = 0.41) and non-targeting sgRNAs (AUC = 0.45), indicating that CRISPRi mitigates cutting-related toxicity.

Previous comparisons between CRISPRko, CRISPRi, and RNAi suggested that the two technologies may identify different biological categories of essential genes[42,43]. To compare CRISPRi to CRISPRko, we examined gene-level correlations between Brunello and Dolcetto in A375 cells (Pearson $R = 0.69$; Fig. 4b). For any gene, differences in essentiality between CRISPRko and CRISPRi may reasonably be attributed to the efficacy of individual sgRNAs rather than gene-intrinsic differences between knockout and knockdown. Therefore, to determine whether there were categories of genes systematically more depleted by one technology, we performed Gene Set Enrichment Analysis[44,45] (GSEA) using the KEGG gene sets. We saw excellent correspondence between Brunello and Dolcetto (Pearson $R = 0.88$; Fig. 4c), indicating that, at least generally, genes manifesting a proliferation defect are equally assessed by both technologies when using optimized reagents.

Interestingly, one gene set, "Systemic Lupus Erythematosus," was an outlier in this comparison. When we compared the performance of each individual gene in this set, we saw that numerous histone genes were essential when assessed by CRISPRko but not by CRISPRi in both A375 and HT29 cells (Fig. 4d; Supplementary Figure 4a; Supplementary Data 4). This observed difference between gene knockout and knockdown may represent a false positive with CRISPRko or a false negative with CRISPRi. A simple explanation is that regions containing histone clusters[46] are copy number amplified and therefore show cutting toxicity with CRISPRko. However, neither region of chromosome 1 or 6 shows evidence of high copy number in A375 or HT29 cells (Fig. 4e; Supplementary Figure 4b). Additionally, several non-histone genes near the histone clusters on chromosome 1 show comparable depletion with Brunello and Dolcetto in A375 cells (Fig. 4f), further suggesting that these regions are neither copy number amplified nor inaccessible to CRISPRi reagents.

Possibly, CRISPRi sgRNAs are less effective at targeting histone genes due to systematic biases in TSS annotation. Another unusual feature of histone genes is that most are only transcribed during S phase[47]. Intriguingly, in A375 cells, two notable exceptions to the differential depletion of histones were *H2AFX* and *H2AFZ*, two histone genes located outside histone clusters and transcribed in a replication-independent fashion. Therefore, CRISPRi could be less effective at repressing gene expression during S phase. Finally, this observation may represent a true differential response to gene knockout and gene knockdown: perhaps low levels of histone gene expression are sufficient for viability. Consistent with this hypothesis, we examined CRISPRko and RNAi datasets from the DepMap[26,48–50] and observed that

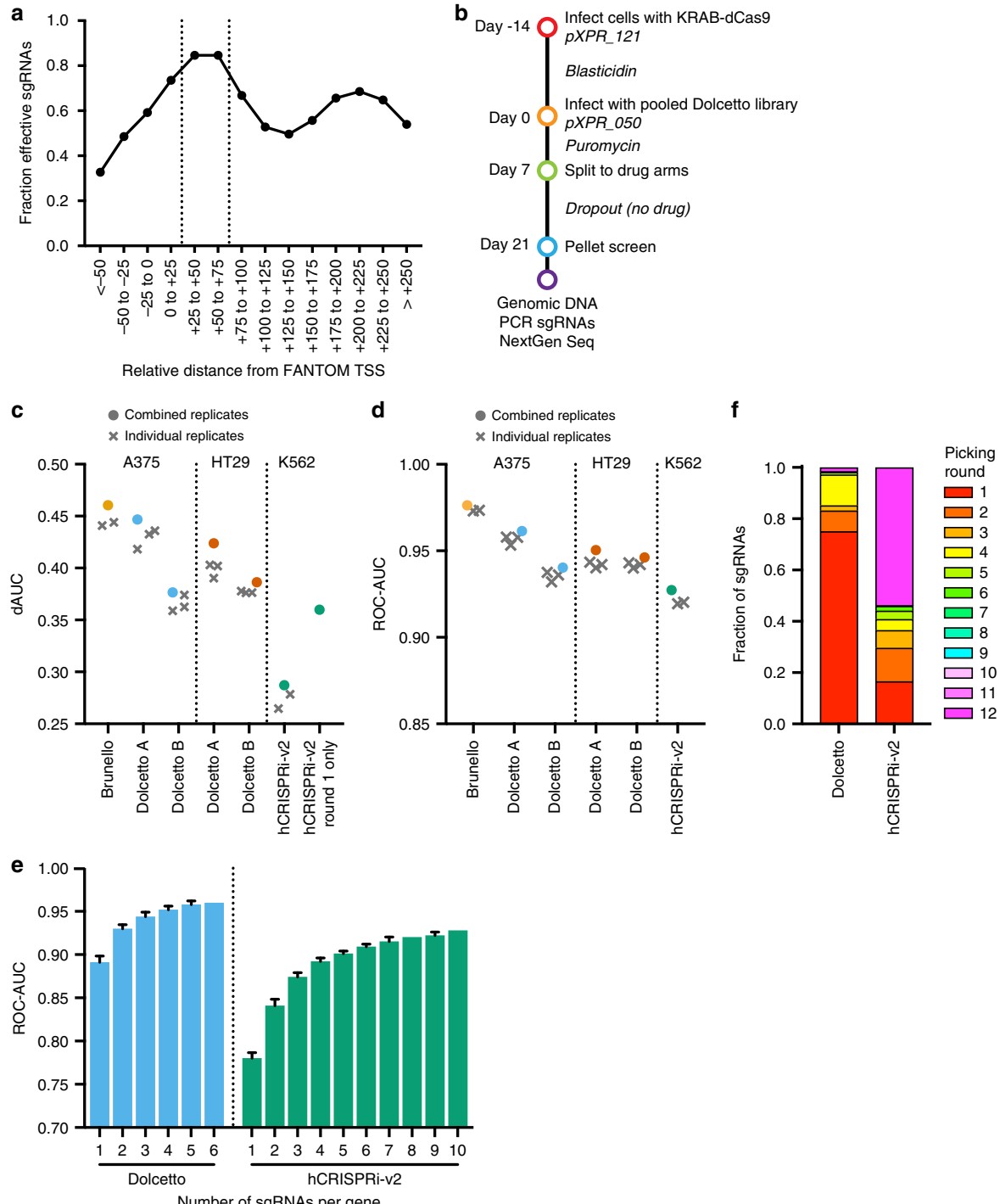

**Fig. 3** CRISPRi screening with Dolcetto. **a** Comparison of sgRNA activity as a function of distance from the FANTOM-annotated transcription start site (TSS). Vertical dotted lines indicate the preferred targeting site. **b** Schematic of viability screens performed with Dolcetto in two cell lines. **c** dAUC comparison of sgRNA-level performance across CRISPRi libraries by cell line. dAUC values of individual replicates are plotted as X's while dAUCs calculated using the average log2-fold change of replicates are plotted as circular points. For hCRISPR-v2 Round 1 only, the dAUC of the hCRISPR-v2 sgRNAs that would be picked in round 1 of our selection heuristic is plotted. Brunello dAUC values are plotted for comparison. **d** ROC-AUC comparison of gene-level performance across CRISPRi libraries by cell line. ROC-AUC values of individual replicates are plotted as X's while ROC-AUCs calculated using the average log2-fold change of replicates are plotted as circular points. Brunello ROC-AUC values are plotted for comparison. **e** Subsampling analysis, calculating the ROC-AUC of n sgRNAs per gene in Dolcetto and hCRISPRi-v2. Error bars represent s.d. calculated from different iterations of library sampling. **f** Fraction of sgRNAs selected in each round using the heuristics described in Supplementary Table 2 for Dolcetto and hCRISPRi-v2

histone genes show viability effects in the former but not the latter across hundreds of cell lines (Fig. 4g). However, as essential genes are also differentially depleted with RNAi and CRISPRko, we cannot rule out that the differential depletion of histone genes simply reflects the overall ineffectiveness of RNAi reagents. These possibilities highlight that, despite the overall similar performance of Brunello and Dolcetto, observed differences may shed light on interesting biological phenomena.

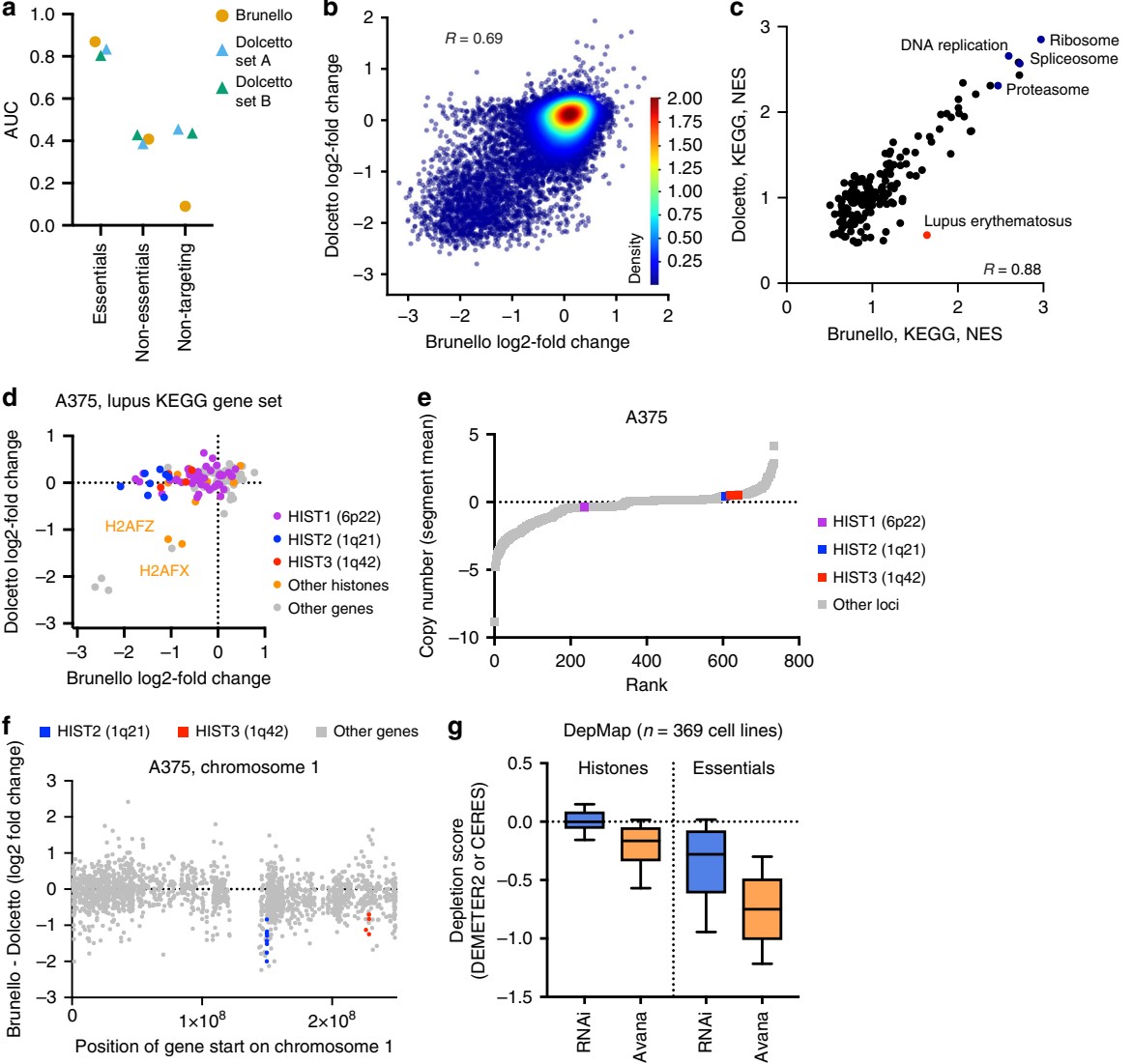

**Fig. 4** Comparison of CRISPRko and CRISPRi. **a** Comparison of AUCs for essential, non-essential and non-targeting sgRNAs across CRISPRko and CRISPRi libraries screened in A375 cells. CRISPRko is plotted as circular points and CRISPRi is plotted as triangles. **b** Comparison of the log2-fold change of genes in Brunello and Dolcetto (average of Sets A and B). Pearson correlation is reported. **c** Scatter plot comparing the GSEA normalized enrichment scores (NES) for KEGG genes sets for Brunello and Dolcetto. Selected gene sets are annotated. **d** Scatter plot comparing log2-fold change of genes from the KEGG Systemic Lupus Erythematosus gene set for Brunello and Dolcetto in A375 cells. Histone genes are colored based on their chromosomal location. **e** Segmented copy number (segment mean; log2-fold change from average) of genetic loci in A375 cells. **f** Position of gene start along chromosome 1 compared to the difference in log2-fold change between Brunello and Dolcetto in A375 cells. **g** Depletion of histone genes and essential genes in data from the Project Achilles screens in 369 cell lines screened with both RNAi and CRISPR (Avana library). The box represents the 25th, 50th, and 75th percentiles; whiskers show 10th and 90th percentiles

**Optimized genome-wide CRISPRa library**. The use of dCas9 also enables transcriptional activation (CRISPRa). Here, multiple strategies to recruit transcriptional machinery have proven effective[51], including directly fusing activation domains to dCas9 (e.g. VP16), the use of the "Sun Tag" to recruit multiple copies of VP16[52], or modifying the tracrRNA region to include structured RNA domains such as MS2 that can recruit additional transcription factors[20]. Based on previous studies[53], we introduced two MS2 and two PP7 stem loops to create tracr-v14, a design that may allow higher-order combinations of domain recruitment (Fig. 5a). Because we encountered difficulties generating lentivirus of reasonable titer from the MS2-p65-HSF1 construct described in the SAM system[20], we instead added a PP7-p65-HSF1 cassette to the library vector to create a two-vector CRISPRa screening system.

As with CRISPRi, sgRNA location is essential for effective gene upregulation. We again used FANTOM to annotate the TSS, but instead targeted a window that was 150–75 nucleotides upstream of the TSS, based on re-analysis of previous data[18,40] (Fig. 5b). We successively relaxed location, on-target sequence score, and potential off-targets to select the six best sgRNAs for each gene (Supplementary Table 4), which were divided into Set A and Set B. This library, named Calabrese, was cloned into the pXPR_502 library vector, which contains tracr-v14 as well as the transcriptional activation domains p65-HSF1.

Unlike CRISPRko and CRISPRi, CRISPRa lacks an obvious gold standard gene set with which to assess screen performance and compare previously published screens (Supplementary Table 5). Therefore, to assess performance, we performed a vemurafenib-resistance screen similar to that previously executed

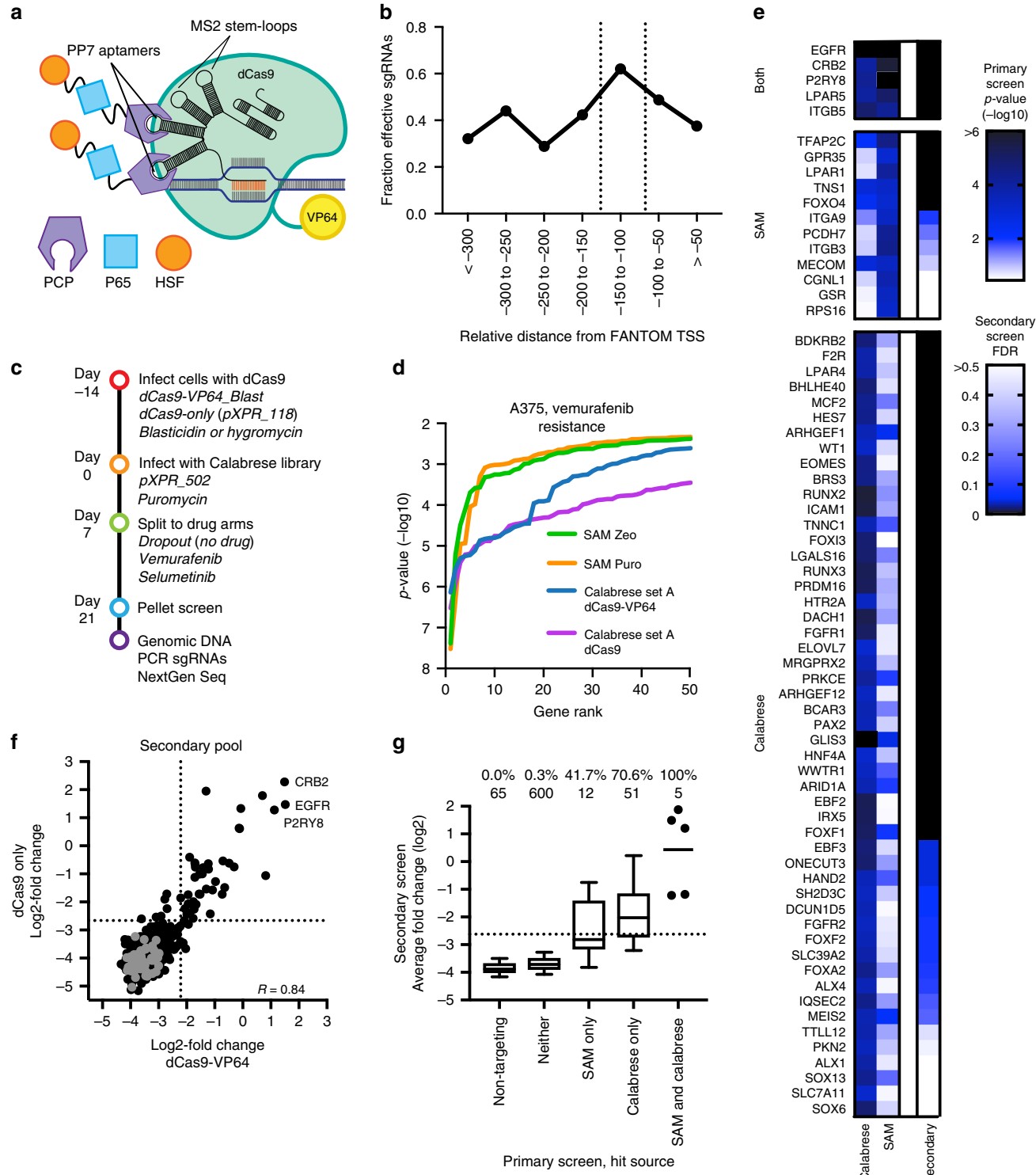

**Fig. 5** CRISPRa screening with Calabrese. **a** Schematic of CRISPRa components. **b** Comparison of sgRNA activity as a function of distance from annotated transcription start site (TSS). Vertical dotted lines indicate the preferred targeting window. **c** Schematic of screens performed with Calabrese. **d** Distribution of the p-values calculated using a hypergeometric distribution equivalent to a one-sided Fisher's exact test for the top 50 genes in vemurafenib resistance screens in A375 cells with the SAM and Calabrese libraries. **e** Comparison of the primary screen p-values calculated as in **d** and secondary screen false discovery rates (FDR). All genes shown scored with p-value < 10⁻³ in either the Calabrese or SAM primary screen, and are grouped accordingly. **f** Comparison of the average log2-fold change values for all sgRNAs targeting a gene for each of two dCas9 constructs screened with a secondary pool. Sets of non-targeting control sgRNAs are gray; dotted lines represent a 5% FDR cutoff. The gene-level Pearson correlation is indicated. **g** Validation rate for genes grouped by the primary screen(s) in which they scored. Dotted line indicates FDR <5%. The fraction of genes above this cutoff is reported, as well as the total number of genes in each category. The box represents the 25th, 50th, and 75th percentiles; whiskers show 10th and 90th percentiles. For the category with 5 genes, individual values are plotted and the mean indicated

with the SAM system[20], in which a library with 3 sgRNAs per gene was screened in duplicate in two vectors, which conferred either zeocin or puromycin resistance. We screened Calabrese Set A in both A375 cells stably expressing dCas9-VP64[5,6,20] and A375 cells stably expressing dCas9 without the VP64 domain (Fig. 5c; Supplementary Data 5); in the latter, activation domains were solely recruited via the PP7 stem loops in the library vector. Comparing the log2-fold change of vemurafenib-treated to untreated cells showed replicate correlation of 0.08 (dCas9-VP64) and 0.47 (dCas9 only). In comparison, the replicate correlation for SAM screens with zeocin and puromycin resistance was 0.04 and 0.24, respectively. Low replicate correlation is generally expected in positive selection screens, as the majority of sgRNAs do not confer resistance.

After averaging together sgRNAs targeting the same gene, we found that both Calabrese screens revealed substantially more hits at various p-value thresholds than either screen with the SAM library (Fig. 5d), indicating better concordance of sgRNAs targeting the same gene. For example, at a p-value cut-off of $10^{-4}$, the Calabrese screens with dCas9-VP64 and dCas9 identified 17 and 27 genes, respectively, whereas the two SAM vector screens identified 4 and 5 genes. Notably, both the original and updated[54] SAM library used RefSeq for annotating TSS and did not incorporate on-target activity rules, which may partly explain its performance.

A strong hit with both the Calabrese and SAM libraries was *EGFR* (Fig. 5e), whose activation has previously been identified as a mediator of vemurafenib resistance[55,56]. Four other genes scored in all primary screens: *P2RY8*, *ITGB5*, *LPAR5*, and *CRB2*. *LPAR5* belongs to the lysophosphatidic acid receptor family, which are G-protein coupled receptors (GPCR) that activate the PI3K-AKT signaling pathway[57]; other LPA receptors (*LPAR1*, *LPAR4*) also scored strongly in at least one screen, and *P2RY8* is also a GPCR. To validate novel genes identified in this screen, we created a follow-up library, selecting 72 genes that scored in either or both the SAM and Calabrese screens, as well as 600 genes that did not score in any screens. The library included ~15 sgRNAs per gene, along with 1000 non-targeting controls, and was screened at high coverage in triplicate according to the same timeline as the primary screen (Fig. 5c, Supplementary Data 6). We did not observe substantial differences between the use of dCas9-VP64 or dCas9-only (Fig. 5f), suggesting that, at least in this cell type, the p65-HSF1 transactivator is sufficient for functionally relevant levels of overexpression.

To examine the validation rate of hits from the primary screen, we calculated an empirical false discovery rate (FDR) (see Methods). All 5 of the genes that scored with both libraries in the primary screens validated at an FDR threshold of <5%, as well as 70.6% of the genes that were nominated by the primary screen conducted with the Calabrese library (Fig. 5g). Several genes that scored in the primary screen with the SAM library but not with the Calabrese library, such as *GPR35* and *LPAR1*, validated upon rescreen, suggesting that they were false negatives in the primary Calabrese screen not because of intrinsic differences between the screening systems, such as the use of PP7 or MS2 stem loops, but rather because of the sgRNAs included in each library.

Overall, the secondary screen validated 47 genes at an empirical FDR <5%, including 37 genes that are newly identified by the Calabrese library (Fig. 5e). This set includes numerous transcription factors, such as *WT1*, *HAND2*, *PAX2*, *RUNX2*, *RUNX3*, *EBF2*, and *EBF3*. Several classes of receptors scored, including the receptor tyrosine kinase *FGFR1* (additionally, *FGFR2* had a FDR of 7% in the secondary screen); previously, the growth factor *FGR* was found to confer resistance to BRAF inhibition in this model in a focused ORF overexpression

screen[58]. Additional receptors include the GPCR *F2R* (also known as *PAR-1*), which has previously been implicated in melanoma progression[59], as well as *BRS3*, *HTR2A*, and *BDKRB2*. Further work will be needed to ensure that the resistance phenotypes observed are in fact due to overexpression of the protein-coding gene, rather than, for example, a nearby non-coding transcript, as well as to understand the resistance mechanisms. In sum, the Calabrese library identified both previously validated and novel loci that confer resistance to vemurafenib upon overexpression.

We next tested the performance of Calabrese in another cell line, MelJuSo, a *BRAF*-wildtype, *NRAS*-mutant melanoma line engineered to express dCas9-VP64. We performed a positive selection screen as before (Fig. 5c) with selumetinib, a MEK inhibitor, with both Set A and B in duplicate (Supplementary Data 5). Compared to the no treatment control arm, pairwise Pearson correlation of log2-fold change values across biological replicates was 0.23 for Set A and 0.41 for Set B; after averaging together the three sgRNAs for each gene, the Pearson correlation comparing Set A to Set B was 0.12 (Fig. 6a). Each screen identified several of the same top hit genes that conferred selumetinib resistance upon activation, including the antioxidant responsive transcription factor *NFE2L2* and the multi-drug transporter *ABCB1*; conversely, activation of *DUSP9*, a MAPK phosphatase, sensitized cells to selumetinib. Sets A and B uncovered comparable numbers of genes at various p-value thresholds, and when Sets A and B were combined so that each gene was targeted with 6 sgRNAs, substantially more genes were uncovered at each statistical threshold (Fig. 6b). Thus, both sets of the Calabrese library were able to successfully identify genes conferring resistance to MAPK pathway inhibition.

**Comparison of CRISPRa to ORF overexpression.** A previously executed screen for resistance to MEK inhibition allowed us to compare the activity of a pooled open reading frame (ORF) overexpression library to CRISPRa[60,61]. This ORF screen was also performed in MelJuSo cells, although it used the MEK inhibitor trametinib rather than selumetinib. In that screen, across four replicates, the pairwise correlations ranged from 0.18 to 0.20. Considering the 11,960 genes shared in common across the two libraries, the correlation between the two screens at the gene level was modest (Fig. 6c, Pearson $R = 0.09$). However, common hits emerged. Of the top 100 resistance genes identified by each screen, 12 were found in both screens, including known MAPK oncogenes *EGFR*, *RAF1*, *HRAS*, and *ERBB2*, a statistically significant overlap (p-value $3 \times 10^{-11}$, two-tailed Fisher's exact test).

The differences between the ORF and CRISPRa primary screens could be due to numerous parameters, including differences between selumetinib and trametinib, ineffective sgRNAs, mutations in the ORF constructs, and technical noise. To explore these differences, we constructed a focused secondary CRISPRa pool targeting all genes that scored in the top 100 of the primary screen with either the ORF or CRISPRa library, as well as 337 genes that scored in neither screen; genes were targeted with 12 sgRNAs each, and 419 non-targeting control sgRNAs were added, for a total library of 6700 sgRNAs. This library was screened at high coverage in triplicate in MelJuSo cells via the same time course as the primary screen, with selumetinib, trametinib, and no treatment arms (Supplementary Data 7). Pairwise replicate correlations of log2-fold change values ranged from 0.41 to 0.68, an increase in reproducibility from the primary screens.

Because the primary ORF and CRISPRa screens used different small molecules, we first compared the results of these two treatments and observed that selumetinib and trametinib were

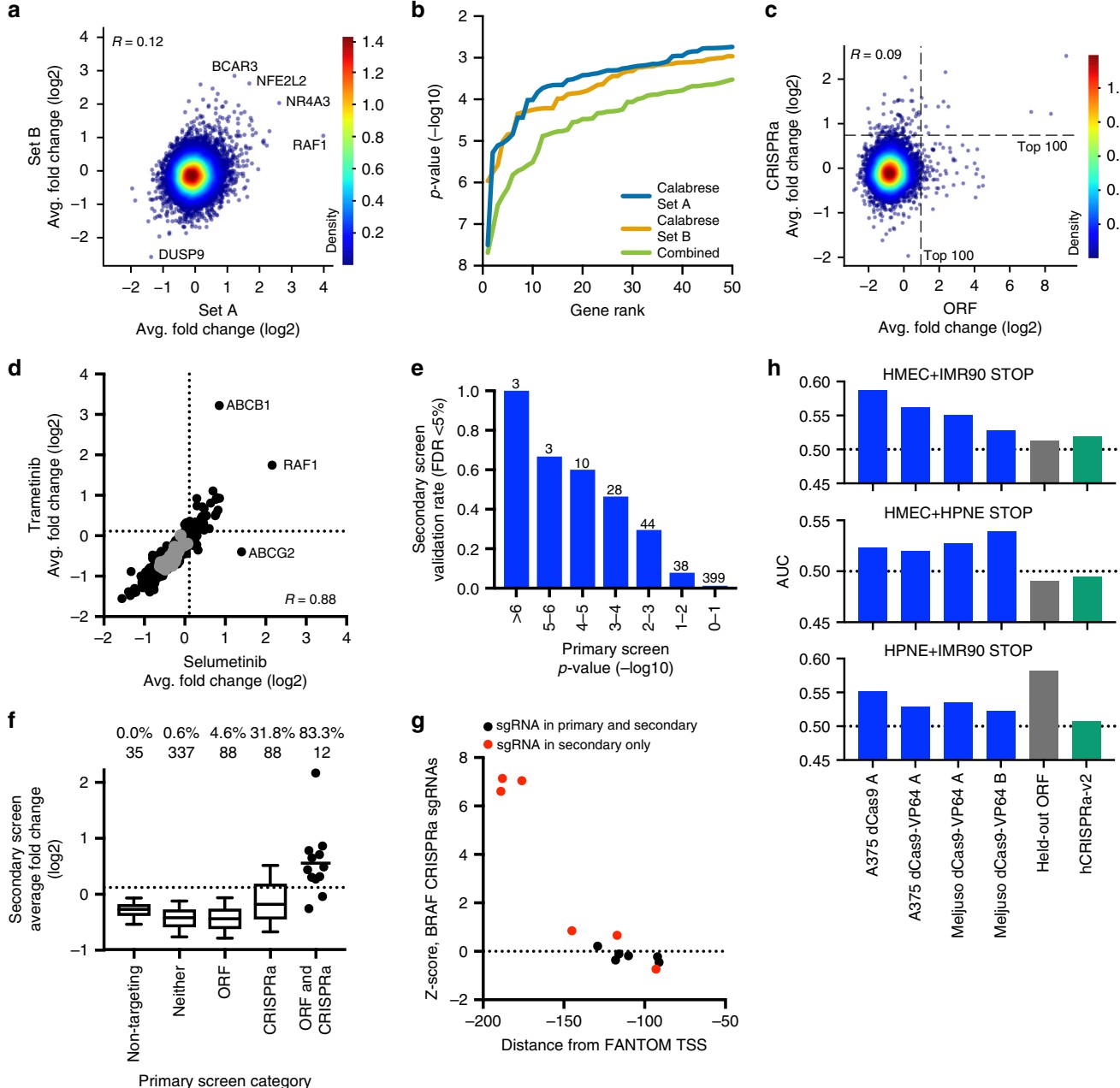

**Fig. 6** Comparison of CRISPRa and ORF technology. **a** Comparison of log2-fold change values for Calabrese Set A and Set B. Each dot represents the average of the 3 sgRNAs per gene. Top scoring genes are annotated. **b** Distribution of the *p*-values for the top 50 genes in each set of Calabrese, as well as both sets combined, for selumetinib resistance screens in MelJuSo cells. *P*-values were calculated using a hypergeometric distribution equivalent to a one-sided Fisher's exact test. **c** Comparison of log2-fold change for MEK inhibition screens in MelJuSo cells, for an ORF library screened with trametinib and the Calabrese library screened with selumetinib. Dotted lines indicate the top 100 genes in each screen. **d** Comparison of the average log2-fold change values for all sgRNAs targeting a gene in the secondary pool screened with either selumetinib or trametinib. Sets of non-targeting control sgRNAs are colored in gray; dotted lines represent a 5% FDR cutoff. **e** Validation rate of genes in the secondary screen grouped by their *p*-value in the primary screen (combined sets; see (**b**)) for selumetinib resistance in MelJuSo cells. Number of genes per category is indicated. **f** Validation rate for genes grouped by the primary screen in which they scored. Dotted line indicates FDR <5%. The fraction of genes pass this cutoff is reported, as well as the total number of genes in each category. The box represents the 25th, 50th, and 75th percentiles; whiskers show 10th and 90th percentiles. For the category with 12 genes, the individual values are plotted and the mean indicated. **g** Secondary screen results for sgRNAs targeting *BRAF*. Black points are sgRNAs present in both the primary (Calabrese) and secondary pools; red points are sgRNAs that were only included in the secondary pool. **h** Area under the curve calculated for STOP genes in CRISPRa and held-out ORF screens. For each plot, a different ORF screen cell line (HMEC, IMR90, or HPNE) was held out, and the other two, indicated at the top of each graph, were used to generate a list of STOP genes

well correlated (*R* = 0.88, Fig. 6d). Interestingly, the two major outliers were multi-drug transporters, with *ABCG2* scoring only with selumetinib and *ABCB1* scoring more strongly with trametinib. These results argue that the differences observed between the primary screens were not largely driven by the choice of small molecule.

As with the vemurafenib secondary pool, we calculated a validation rate based on an empirical FDR threshold of <5% and

observed a clear trend in the validation rates of genes in the secondary screen based on their performance in the primary screen (Fig. 6e). We next examined the validation rate of genes nominated by each primary screen. In the selumetinib resistance screen, 10 of the 12 genes that scored via both technologies validated at an FDR <5%, as did 31.8% of the CRISPRa hits from the primary screen (Fig. 6f). Only a small fraction (4.8%) of genes that originally scored only as ORFs validated upon the inclusion of additional sgRNAs, suggesting that these differences are not simply false negatives arising from technical noise in the primary CRISPRa screen. One gene that validated, *KSR1*, has recently been shown to activate BRAF via interactions with MEK[62], as did *BRAF* itself.

Closer examination of the sgRNAs targeting *BRAF* revealed that all of the sgRNAs that were in the original Calabrese library continued not to provide resistance to MEK inhibition, reproducing the negative result in the primary screen (Fig. 6g). However, three sgRNAs that were newly added to the secondary pool scored strongly, with ranks 5, 6, and 7 out of 6700 in the secondary pool. These three sgRNAs are located farther from the FANTOM-annotated transcription start for this gene, potentially indicating an alternative TSS for *BRAF* in this cell type; indeed, the annotated p1 TSS constituted only 58% of the *BRAF* CAGE peak reads, strengthening this hypothesis. Thus, as a general strategy for CRISPRa validation of ORF hits, it may be more informative to emphasize greater spacing of sgRNAs across the annotated transcription start site(s) for a gene, in order to capture genes with multiple or inaccurately annotated TSSs.

A limitation of drug resistance screens for benchmarking library performance is the lack of an orthogonal gold standard. We therefore sought to assess the effectiveness of Calabrese at modulating proliferation-related genes in the absence of drug treatment. A recent study using a large-scale ORF library of ~10,000 clones screened in 3 cancer cell lines identified growth inhibitory (STOP) and promoting (GO) genes[63]. Most STOP and GO genes were identified as cell-type specific, with only 103 and 3 genes, respectively, scoring in all 3 lines. Nevertheless, these data are an unbiased comparator for the performance of CRISPRa libraries. We generated 3 lists of STOP genes by holding out one cell line and requiring that the gene score in both of the remaining two cell lines. We then determined the AUC for the held-out ORF screen and the no-treatment arms of the Calabrese screens described above; we also analyzed the hCRISPRa-v2 screen performed in K562 cells[19]. Although the AUC values were modest, likely due to the cell-type specificity of most STOP genes, in all cases the AUC was >0.5 for the screens performed with Calabrese, indicating preferential depletion of STOP genes. Further, for two of the three lists of STOP genes, the Calabrese CRISPRa screens had a higher AUC than the held-out ORF screen, and in all cases outperformed the hCRISPRa-v2 library (Fig. 6h). That sgRNAs activating growth inhibitory genes are preferentially depleted suggests that Calabrese is effective at modulating proliferation.

## Discussion

Here we present three compact human genome-wide libraries for CRISPRko, CRISPRi, and CRISPRa and test their performance via 14 genome-wide screens conducted across 3 cell types. We demonstrate that these libraries outperform larger libraries on both the gene and sgRNA level, which we attribute to improved sgRNA design. We also expect that the mouse versions of the CRISPRko, CRISPRa, and CRISPRi libraries, named Brie, Caprano, and Dolomiti, respectively, will also offer improved performance, as previous studies did not observe significant differences between sgRNA design criteria in mouse and human cells[9,12].

Our results highlight the utility of modifying gene expression in multiple ways to fully probe gene function. We demonstrate that Brunello and Dolcetto, for CRISPRko and CRISPRi, respectively, are both highly effective loss-of-function libraries, with comparable depletion of essential genes. We attribute this performance to improvements in sgRNA design, and demonstrate that application of our CRISPRko or CRISPRi sgRNA selection schemes to other published libraries improves their discrimination of essential and non-essential genes. Overall, we find a strong correlation between the gene sets depleted by Brunello and Dolcetto, suggesting that optimized reagents may overcome some of the observed systematic differences between gene knockout by CRISPR and gene knockdown by RNAi[42]. Still, CRISPRko and CRISPRi each have distinct strengths and weaknesses. CRISPRko leads to cutting-related toxicity at copy number amplified loci[24], and, because the gene dosage is complete knockout in many cases, may result in false negatives in positive selection screens when a gene is required for viability at some level but shows a phenotype on partial inhibition. CRISPRi offers the possibility of titrating the amount of gene expression, but may downregulate multiple genes at bidirectional promoters[41]. Therefore, CRISPRko and CRISPRi screens represent valuable orthogonal approaches to separate technical artifacts from true hits.

Upregulation of genes through CRISPRa likewise represents a complementary method for pooled screening that can reveal the function of lowly-expressed genes and pathways more effectively modulated by gene activation. When we compared CRISPRa and ORF screens for resistance to MEK inhibitors, we found that the technologies identified a number of common top hits, but also numerous unique ones; the large majority of these novel genes identified by CRISPRa validated in a secondary screen. Both technologies have sources of false negatives and positives that may explain these differential hits. Sources of false negatives for ORF technology include overexpression of an irrelevant splice isoform; for CRISPRa, false negatives may arise when the target gene is not effectively overexpressed, due to poor sgRNA design, inaccurate TSS annotation, or inaccessible chromatin environment. False positives can occur in ORF screens when an ORF is overexpressed to a level never achieved by the cell endogenously and, in CRISPRa screens, when multiple genes are upregulated at bidirectional promoters. Therefore, both ORF and CRISPRa screens are valuable and complementary elements of the pooled screening toolbox.

Overall, we expect that these optimized picking criteria for CRISPRi and CRISPRa sgRNAs will be broadly useful and have made our sgRNA design tool publically available (broad.io/gpp-sgrna-design). Moreover, although the libraries introduced here target coding genes, we expect that our picking criteria could be adapted to design highly active sgRNAs for large paired deletions[64], modulation of lncRNAs, or other applications that require effective Cas9 recruitment.

Although CRISPR-Cas9 technology has greatly expanded the capacity to conduct large-scale, unbiased screens, designing screenable model systems remains a challenge; often, primary cells or in vivo models best capture relevant disease biology, but are limited in cell number and therefore library size. Once a high-quality, screenable model system has been developed, it is therefore advantageous to apply multiple technologies to uncover distinct hits. The compact CRISPRko, CRISPRi, and CRISPRa libraries introduced here will enable genome-scale interrogation of gene function in diverse model systems and assist in expanding pooled screening beyond easily cultured cell lines.

## Methods

**Vectors**. The following vectors were used in this study and are available to the academic community on Addgene:

pLX_311-Cas9: SV40 promoter expresses blasticidin resistance; EF1a promoter expresses SpCas9 (Addgene 96924).

dCAS-VP64_Blast (pXPR_109): EF1a promoter expresses dSpCas9-VP64 and 2A site provides blasticidin resistance (Addgene 61425).

pXPR_118: EF1a promoter expresses dSpCas9 and 2A site provides hygromycin resistance (Addgene 113667).

pXPR_121: SV40 promoter expresses blasticidin resistance; EF1a promoter expresses KRAB-dSpCas9 (Addgene 96918).

lentiGuide-Puro (pXPR_003): EF1a promoter expresses puromycin resistance; U6 promoter expresses customizable sgRNA element (Addgene 52963).

lentiCRISPRv2 (pXPR_023): EF1a promoter expresses SpCas9 and 2A site provides puromycin resistance; U6 promoter expresses customizable sgRNA element (Addgene 52961).

pXPR_050: EF1a promoter expresses puromycin resistance; U6 promoter expresses customizable sgRNA element with tracr-v2 (Addgene 96925).

pXPR_502: PGK promoter expresses PP7 tethered to P65-HSF1 and 2A site provides puromycin resistance; U6 promoter expresses customizable sgRNA element with tracr-v14, containing 2 MS2 and 2 PP7 stem loops (Addgene 96923)

Brunello library: Addgene 73178 (in lentiGuide-Puro) and Addgene 73179 (in lentiCRISPRv2)

Calabrese library: Addgene 92379 (Set A), 92380 (Set B)

Dolcetto library: Addgene 92385 (Set A), 92386 (Set B)

**CRISPRi and CRISPRa library design**. To design the CRISPRa and CRISPRi libraries, we first used RefSeq to define a list of protein-coding genes. Next, to annotate the FANTOM5-defined TSS for each gene, we used v1 of the reprocessed FANTOM5 CAGE data (accessed March 2016), in which the CAGE peaks from phase 1 and phase 2 were remapped to current genome assemblies (hg38 and mm10) using the liftOver tool[65,66]. Using the annotation file found in the reprocessed FANTOM data, we took the "CAGE Peak ID" for the highest ranked peak of every annotated gene in the FANTOM5 dataset, irrespective of the distance from the peak to Ensembl transcripts. We then searched for each CAGE Peak ID in the corresponding bed file; the position marked as the "start of the representative TSS position" was considered the TSS for that gene. For the protein-coding genes that did not have TSS information in FANTOM5, we used the TSS of the principal transcript from Ensembl. If the TSS was not found in Ensembl, the TSS from NCBI was used.

For each gene, we first designed all possible sgRNAs with NGG PAM in the window of −300 to +300 relative to the TSS. We then calculated the Rule Set 2 scores for these sgRNAs and annotated each sgRNA with the number of perfectly matched off-target sites in the human genome.

We then picked sgRNAs based on their position relative to the TSS, number of off-target matches, and on-target score to fill a quota of six sgRNAs per gene (Supplementary Table 2 for CRISPRi and Supplementary Table 4 for CRISPRa). One or more of these three criteria were relaxed in each picking round until the quota was filled. A standard set of 992 non-targeting controls was added to each library. We then divided each library into Set A and Set B with the former having the top 3 sgRNAs and the latter having the next 3 sgRNAs per gene.

**Library production**. Oligonucleotide pools were synthesized by CustomArray. BsmBI recognition sites were appended to each sgRNA sequence along with the appropriate overhang sequences (bold italic) for cloning into the sgRNA expression plasmids, as well as primer sites to allow differential amplification of subsets from the same synthesis pool. The final oligonucleotide sequence was thus: 5′-[Forward Primer]CGTCTCA*CACC*G[sgRNA, 20 nt]*GTTT*CGAGACG[Reverse Primer].

Primers were used to amplify individual subpools using 25 μL 2x NEBnext PCR master mix (New England Biolabs), 2 μL of oligonucleotide pool (~40 ng), 5 μL of primer mix at a final concentration of 0.5 μM, and 18 μL water. PCR cycling conditions: 30 s at 98 °C, 30 s at 53 °C, 30 s at 72 °C, for 24 cycles. In cases where a library was divided into subsets unique primers could be used for amplification:

Primer Set; Forward Primer, 5′ – 3′; Reverse Primer, 5′ – 3′

1; AGGCACTTGCTCGTACGACG; ATGTGGGCCCGGCCACCTTAA

2; GTGTAACCCGTAGGGCACCT; GTCGAGAGCAGTCCTTCGAC

3; CAGCGCCAATGGGCTTTCGA; AGCCGCTTAAGAGCCTGTCG

4; CTACAGGTACCGGTCCTGAG; GTACCTAGCGTGACGATCCG

5; CATGTTGCCCTGAGGCACAG; CCGTTAGGTCCCGAAAGGCT

6; GGTCGTCGCATCACAATGCG; TCTCGAGCGCCAATGTGACG

The resulting amplicons were PCR-purified (Qiagen) and cloned into the library vector via Golden Gate cloning with Esp3I (Fisher Scientific) and T7 ligase (Epizyme); the library vector was pre-digested with BsmBI (New England Biolabs). The ligation product was isopropanol precipitated and electroporated into Stbl4 electrocompetent cells (Life Technologies) and grown at 30 °C for 16 h on agar with 100 μg mL⁻¹ carbenicillin. Colonies were scraped and plasmid DNA (pDNA) was prepared (HiSpeed Plasmid Maxi, Qiagen). To confirm library representation and distribution, the pDNA was sequenced.

**Virus production**. For small-scale virus production, the following procedure was used: 24 h before transfection, HEK293T cells were seeded in 6-well dishes at a density of $1.5 \times 10^6$ cells per well in 2 mL of DMEM + 10% FBS. Transfection was performed using TransIT-LT1 (Mirus) transfection reagent according to the manufacturer's protocol. Briefly, one solution of Opti-MEM (Corning, 66.25 μL) and LT1 (8.75 μL) was combined with a DNA mixture of the packaging plasmid pCMV_VSVG (Addgene 8454, 250 ng), psPAX2 (Addgene 12260, 1250 ng), and the transfer vector (e.g., pLentiGuide, 1250 ng). The solutions were incubated at room temperature for 20–30 min, during which time media was changed on the HEK293T cells. After this incubation, the transfection mixture was added dropwise to the surface of the HEK293T cells, and the plates were centrifuged at 1000 g for 30 min at room temperature. Following centrifugation, plates were transferred to a 37 °C incubator for 6–8 h, after which the media was removed and replaced with DMEM +10% FBS media supplemented with 1% BSA.

A larger-scale procedure was used for pooled library production. 24 h before transfection, $18 \times 10^6$ HEK293T cells were seeded in a 175 cm² tissue culture flask and the transfection was performed the same as for small-scale production using 6 mL of Opti-MEM, 305 μL of LT1, and a DNA mixture of pCMV_VSVG (5 μg), psPAX2 (50 μg), and 40 μg of the transfer vector. Flasks were transferred to a 37 °C incubator for 6–8 h; after this, the media was aspirated and replaced with BSA-supplemented media. Virus was harvested 36 h after this media change.

**Cell culture**. A375, HT29, and MelJuSo cells were obtained from the Cancer Cell Line Encyclopedia. HEK293Ts were obtained from ATCC (CRL-3216).

All cell lines were routinely tested for mycoplasma contamination and were maintained without antibiotics except during screens, when media was supplemented with 1% penicillin/streptomycin. Cell lines were kept in a 37 °C humidity-controlled incubator with 5.0% CO2 and were maintained in exponential phase growth by passaging every 2–3 days.

For each cell line, the following media and doses of polybrene, puromycin, blasticidin, and hygromycin, respectively, were used:

A375: RPMI + 10% FBS; 1 μg mL⁻¹ (0.5 μg mL⁻¹ for no-spin transductions); 1 μg mL⁻¹; 5 μg mL⁻¹; 50 μg mL⁻¹

HT29: DMEM + 10% FBS; 1 μg mL⁻¹; 2 μg mL⁻¹; 5 μg mL⁻¹; N/A

MelJuSo: RPMI + 10% FBS; 4 μg mL⁻¹; 1 μg mL⁻¹; 2 μg mL⁻¹; N/A

Vemurafenib (S1267) and selumetinib (S1008) were obtained from Selleckchem and screened at doses of 2 μM and 1.5 μM, respectively.

**Determination of lentiviral titer**. To determine lentiviral titer for spin transductions, cell lines were transduced in 12-well plates with a range of virus volumes (e.g. 0, 150, 300, 500, and 800 μL virus) with $3.0 \times 10^6$ cells per well in the presence of polybrene. The plates were centrifuged at 640 x g for 2 h and then transferred to a 37 °C incubator for 4–6 h. Each well was then trypsinized, and an equal number of cells seeded into each of two wells of a 6-well dish. Two days post-transduction, puromycin was added to one well out of the pair. After 5 days, both wells were counted for viability. A viral dose resulting in 30–50% transduction efficiency, corresponding to an MOI of ~0.35–0.70, was used for subsequent library screening.

To determine lentiviral titer for no-spin transductions, cell lines were seeded in 6-well plates in the presence of polybrene (0.5 μg mL⁻¹) and virus at a range of volumes (e.g. 0, 50, 100, 200, 400, and 600 μL virus), with two wells per virus volume. 16–18 h after seeding, virus-containing media was replaced with fresh media. Two days post-transduction, puromycin was added to one well out of the pair. After 5 days, both wells were counted for viability. A viral dose resulting in 30–50% transduction efficiency, corresponding to an MOI of ~0.35–0.70, was used for subsequent library screening.

**CRISPRko and CRISPRi screens**. Cas9 or dCas9 derivatives were made by transducing with the lentiviral vector pLX_311-Cas9, which expresses blasticidin resistance from the SV40 promoter and Cas9 from the EF1α promoter, or pXPR_121, which expresses blasticidin resistance from the SV40 promoter and KRAB-dCas9 from the EF1α promoter, respectively.

Prior to screening-scale transduction, Cas9 and KRAB-dCas9-expressing cell lines were selected with blasticidin; they were then transduced in two or three biological replicates at a low MOI (~0.5). Transductions were performed with enough cells to achieve a representation of at least 500 cells per sgRNA per replicate, taking into account a 30–50% transduction efficiency. Throughout the screen, cells were split at a density to maintain a representation of at least 500 cells per sgRNA, and cell counts were taken at each passage to monitor growth. Puromycin selection was added 2 days post-transduction and was maintained for 5–7 days. 3 weeks post-transfection, cells were pelleted by centrifugation, resuspended in PBS, and frozen promptly for genomic DNA isolation.

**CRISPRa screens**. Cell lines expressing dCas9-VP64 and dCas9 were made by transducing cells with the lentiviral vector pXPR_109, which expresses blasticidin resistance from a 2A site and dCas9-VP64 from the EF1α promoter, or pXPR_118, which expresses hygromycin resistance from a 2A site and dCas9 from the EF1α promoter, respectively. Prior to screening-scale transduction, dCas9-VP64 and dCas9 cell lines were selected with blasticidin and hygromycin, respectively.

For the dCas9-VP64 screens, cell lines expressing dCas9-VP64 were transduced with the Calabrese library in two biological replicates at a low MOI (~0.5). Transductions were performed with enough cells to achieve a representation of at least 500 cells per sgRNA per replicate, taking into account a 30–50% transduction efficiency. Throughout the screen, cells were split at a density to maintain a representation of at least 500 cells per sgRNA, and cell counts were taken at each passage to monitor growth. Puromycin selection was added 2 days post-transduction and was maintained for 5–7 days. After puromycin selection was complete, each replicate was split to no drug and drug treatment arms, each at a representation of at least 500 cells per sgRNA. A375 screens were performed with 2 μM vemurafenib; MelJuSo screens were performed with 1.5 μM selumetinib. 14 days after the initiation of drug treatment, cells were pelleted by centrifugation, resuspended in PBS, and frozen promptly for genomic DNA isolation.

For the dCas9-only screens, A375 cells expressing dCas9 were transduced with the Calabrese library in two biological replicates at a low MOI (~0.5) via a low-representation, no-spin method. Transductions were performed to achieve a representation of at least 300 sgRNAs per replicate, taking into account a 30–50% transduction efficiency. Cells were seeded into T175 flasks in a total volume of 20 mL of virus-containing media with polybrene at $0.5\,\mu g\,mL^{-1}$. Flasks were then transferred to an incubator overnight. 16–18 h after seeding, the virus-containing media was replaced with fresh media and cells were expanded to achieve a representation of at least 500 transduced cells per sgRNA. Puromycin selection was added 2 days post-transduction and was maintained for 5–7 days. After puromycin selection was complete, each replicate was split to no drug and drug treatment arms, each at a representation of at least 500 cells per sgRNA. 14 days after the initiation of drug treatment, cells were pelleted by centrifugation, resuspended in PBS, and frozen promptly for genomic DNA isolation.

For secondary screens, cells expressing either dCas9 or dCas9-VP64 were transduced with the secondary pool in three biological replicates at a low MOI (~0.5). Transductions were performed with enough cells to achieve a representation of at least 500 cells per sgRNA per replicate, taking into account a 30–50% transduction efficiency. Throughout the screen, cells were split at a density to maintain a representation of at least 2000 cells per sgRNA, and cell counts were taken at each passage to monitor growth. Puromycin selection was added 2 days post-transduction and was maintained for 5–7 days. After puromycin selection was complete, each replicate was split to no drug and drug treatment arms, each at a representation of at least 2000 cells per sgRNA. A375 secondary screens were performed with 2 μM vemurafenib; MelJuSo secondary screens were performed with 10 nM trametinib or 1.5 μM selumetinib. 14 days after the initiation of drug treatment, cells were pelleted by centrifugation, resuspended in PBS, and frozen promptly for genomic DNA isolation.

**Genomic DNA preparation and sequencing.** Genomic DNA (gDNA) was isolated using the QIAamp DNA Blood Maxi (3e7–1e8 cells), Midi (5e6–3e7 cells), or Mini (<5e6 cells) Kits (Qiagen) as per the manufacturer's instructions. The gDNA concentrations were quantitated by UV Spectroscopy (Nanodrop). For PCR amplification, gDNA was divided into 100 μL reactions such that each well had at most 10 μg of gDNA. Per 96 well plate, a mcaster mix consisted of 75 μL ExTaq DNA Polymerase (Clontech), 1000 μL of 10x Ex Taq buffer, 800 μL of dNTP provided with the enzyme, 50 μL of P5 stagger primer mix (stock at 100 μM concentration), and 2075 μL water. Each well consisted of 50 μL gDNA plus water, 40 μL PCR master mix, and 10 μL of a uniquely barcoded P7 primer (stock at 5 μM concentration). PCR cycling conditions: an initial 1 min at 95 °C; followed by 30 s at 94 °C, 30 s at 52.5 °C, 30 s at 72 °C, for 28 cycles; and a final 10 min extension at 72 °C. P5/P7 primers were synthesized at Integrated DNA Technologies (IDT). PCR products were purified with Agencourt AMPure XP SPRI beads according to manufacturer's instructions (Beckman Coulter, A63880). Samples were sequenced on a HiSeq2000 (Illumina), loaded with a 5% spike-in of PhiX DNA.

Reads were counted by first searching for the CACCG sequence in the primary read file that appears in the vector 5′ to all sgRNA inserts. The next 20 nts are the sgRNA insert, which was then mapped to a reference file of all possible sgRNAs present in the library. The read was then assigned to a condition (e.g. a well on the PCR plate) on the basis of the 8nt barcode included in the P7 primer.

**Screen analysis.** Following deconvolution, the resulting matrix of read counts was first normalized to a reads per million within each condition by the following formula: read per sgRNA/total reads per condition×$10^6$. Reads per million was then log2-transformed by first adding one to all values, which is necessary in order to take the log of sgRNAs with zero reads. For each sgRNA, the log2-fold-change from plasmid DNA (pDNA) was then calculated. All reported log2-fold-changes for dropout screens are relative to pDNA; for positive selection screens with small molecules, the log2-fold-change are calculated relative to the dropout arm (i.e. no drug treatment arm).

To determine p-values for genes to evaluate positive selection screens, we calculated the hypergeometric distribution without replacement based on the rank order of the log-fold-change of the perturbations; this is equivalent to a one-sided Fisher's exact test. Non-targeting control sgRNAs were randomly grouped into dummy genes of the same set size as the library under consideration (e.g. 4 control

sgRNAs for Brunello, 3 for Calabrese Set A, 6 for Calabrese Set A and B combined). The SAM screening data described previously[20] was reanalyzed using this approach.

To calculate an empirical false discovery rate for the two CRISPRa secondary screens, we considered the set of true negatives to be the 600 genes that did not score in the primary screen as well as 'dummy' genes created by random groupings of non-targeting sgRNAs; we note that this approach could slightly overestimate the FDR for true positives if any of the 600 non-scoring genes were false negatives in the primary screen, although these should be rare; indeed, overall these two classes of negative controls performed similarly in the secondary screen.

**Gene set enrichment analysis and analysis of lupus gene set.** Gene set enrichment analysis (GSEA) was performed using the KEGG gene sets. In order to further investigate the genes in the Systemic Lupus Erythematosus gene set, we compared the depletion of genes in the gene set by CRISPRi (Dolcetto) to CRISPRko; for A375 cells, we used CRISPRko data from screens with Brunello, whereas for HT29 cells, we used CRISPRko data from Project Achilles, in which the Avana library was screened with 4 sgRNAs per gene. To analyze copy number amplification in A375 and HT29 cells, we obtained segmented copy number data from Project Achilles[23,26,67] (Avana 17Q4; https://portals.broadinstitute.org/achilles). Gene position and karyotype band data were obtained from Ensembl Biomart (Supplementary Data 4).

We obtained the DEMETER2 score dataset (RNAi_D2_combined_gene_dep_scores.csv) and CERES score dataset (Avana 18Q3 public; gene_effect.csv) from the Cancer Dependency Map portal (https://depmap.org/portal/download/). After filtering to include only the genes (n = 14,831) and cell lines (n = 369) that were screened with both RNAi and CRISPR, we calculated the mean depletion scores across all cell lines for each gene in the histone gene set and essential gene set.

**Analysis of STOP & GO genes.** Using Supplementary Table 2B from the report by Sack et al., we identified STOP genes by requiring first that the ORF was screened in all three cell lines (no 'n/a' values). We then held out each cell line at a time, and required that the ORF clone have a negative log2 value (e.g. Column E) and a p-value < $10^{-4}$ (e.g. Column F) in both other cell lines. Using the resulting list of genes as a reference of essential genes, we then calculated the area-under-the-curve for CRISPRa libraries and the held-out cell line from the ORF dataset.

**Statistical analysis.** All Pearson and Welch's t-tests were performed in GraphPad Prism and are reported in the figure legends.

**Code availability.** All custom Python code used for analysis is available on GitHub: https://github.com/mhegde.

**Reporting Summary.** Further information on experimental design is available in the Nature Research Reporting Summary linked to this article.

## Data availability

All screening data generated during this study are provided as Supplementary Data and are uploaded to the Sequence Read Archive under accession code SRP172473. All screening data analyzed from previous publications were accessed directly from those publications. All other data are available upon reasonable request. SRA data can be accessed at https://www.ncbi.nlm.nih.gov/sra/PRJNA508200.

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

## Acknowledgements

We are grateful to be supported by excellent software engineers: Matthew Greene, Doug Alan, Mark Tomko, Adam Brown, and Tom Green; as well as administrative support: Olivia Baré and Samantha Amaral (Genetic Perturbation Platform, Broad Institute). We thank Tamara Mason and the Walk-Up Sequencing Team for high quality and rapid Illumina sequencing (Genomics Platform, Broad Institute); Tikvak Hayes and Cory Johannessen for sharing the trametinib-resistance ORF screening data prior to publication, and Mahmoud Ghandi for helpful discussions on the interpretation of copy

number data (Cancer Program, Broad Institute). We also thank Laura Sack and Steve Elledge (Harvard Medical School) for helpful guidance on the interpretation of their ORF screening data. This work was supported in part by the Functional Genomics Consortium.

## Author contributions

K.R.S., R.E.H., K.F.D., C.S., E.W.V., M.E.S., and A.G. performed experiments. M.H., R.E.H., K.R.S., A.G., F.P., and J.G.D. analyzed results. D.E.R. and J.G.D. directed the study. R.E.H., K.R.S., and J.G.D. wrote the manuscript with input from all other authors.

## Additional information

**Competing interests:** J.G.D. consults for Tango Therapeutics. The remaining authors declare no competing interests.

