## [Peer Review File · Nature Communications]

Reviewers' Comments:

Reviewer #1:

Remarks to the Author:

Sanson, Hanna and colleagues present a comprehensive benchmarking of latest generation of human CRISPR libraries. They find that newest libraries deliver more accurate data with less sgRNAs. Metrics are developed by which to compare library performance. Other analyses demonstrate useful positional preferences for CRISPRi/a sgRNAs. A new tracr sequence is also developed and implemented for improved on-target performance.

Overall this is a key paper, holding a number of highly practical insights for the large and growing CRISPR screening field. Presentation and writing are top class.

Analyses are logical and intelligent.

Here I raise a number of issues which are relatively minor:

1. To what extent are differences observed between the authors' data and historical data (Fig1,3), due to (a) the potency of the libraries used, or (b) the expertise of the laboratory performing the experiments? (a) is assumed throughout the paper, but it would be worth at least raising the possibility of (b).

2. Tracr engineering: One is left slightly puzzled in the "Modifications to the tracrRNA" section. It is unclear firstly what is the motivation for changing the tracr in the first place. The take-home message seems to be that it yields a modest gain in performance, but with some downsides to off-targeting. I would not expect the authors to repeat any experiments, but perhaps to justify more clearly why this work was undertaken and fits in the present paper.

3. Use of FANTOM CAGE to identify TSS: This is one of the biggest doubts I have in the paper.

- Firstly, the authors do not clearly explain exactly which CAGE dataset is used, even in the Methods. There are several different CAGE datasets, with different cutoffs and years of creation. This must be clearly specified.

- Second: CAGE is an experimental method, performed in cells and tissues. Clearly therefore, differential TSS usage will be reflected in differential CAGE clusters between cell types. How does this influence the library designs performed for CRISPRi and CRISPRa libraries? How does this influence differences in apparent performance for the cell types shown in Fig3c,d? Specifically, it could be that HT29 TSS are simply worse reflected in the TSS dataset, compared to A375.

- Third: It is also unclear in the Methods, how exactly FANTOM TSS were associated with gene TSS. In "CRISPRi and CRISPRa library design" it says that "we picked the TSS with the highest ranked peak..." - but in which length window? And what about the many genes with multiple TSS? I would invite the authors to much more clearly explain these crucial aspects of their method.

- Fourth: This will influence the library design criteria. This should at least be much more clearly explained to the readers in the Discussion.

4. How TSS confidence influences knockdown: A question that is not addressed in this paper, is whether poor TSS annotation could explain poor knockdown in CRISPR screens. Of course it is up to the authors to decide whether to perform this or not. But I think the paper would improve, if some analysis were performed that broke down gene sets according to TSS confidence (ie how confidently the TSS is identified by CAGE), and the % effectiveness of its sgRNAs.

5. Figure 4A, the odd performance of Brunello in Non-Targeting genes: Could the authors offer any explanation for this strange result?

6. p12 / Figure 4 - discrepancy in CRISPRi / KO behaviour for histone genes: This is an interesting observation. Could any light be shed on it by studying other screen data, eg shRNA or siRNA

datasets?

7. Figure 4B: better as a coloured density/intensity plot.

8. Figure 4C: axes are unclear, not sure what is NES.

9. p.9 CAGE stands for "Cap Analysis of Gene Expression"

10. The authors might also briefly mention how results of this paper might impact other related fields, such as screens of long noncoding RNAs, and paired CRISPR deletion screens.

Rory Johnson

Reviewer #2:

Remarks to the Author:

In the manuscript entitled "Up, down, and out: optimized libraries for CRISPRa, CRISPRi, and CRISPR-knockout genetic screens", Dr. Doench and Co-workers present an enormous amount of work including the generation of several genome wide sgRNA libraries against both human and mouse as well as 16 genome wide screens.

It is without a doubt that these libraries, already made accessible via addgene, will (continue to) receive great interest in the scientific community and become a widespread tool for genome wide screens. For those reasons it is also of great interest to systematically compare library performance based on genome wide screens vis a vis of technology (CRISPRa, CRISPRi, CRISPR ko), sgRNA design, and screen setup making this manuscript interesting and relevant in principle. The following section provides suggestions how to improve the comparison and clarity of the presented libraries.

1) The presented manuscript evaluates libraries based on performance of known positive (and negative) controls previously identified in similar sgRNA based screens. Like the control gene set itself, also the sgRNA design algorithm is based on previous screens and thus the used machine learning algorithms will bias towards known genes as well as specific guides depleting in CRISPR screens, thereby resulting in a somewhat circular argument and a self-fulfilling prophecy. It is thus pivotal to highlight this fact as a limitation of the method.

2) The manuscript compares many screens, old and new, performed by the authors as well as others, and derives values to show that e.g. the essentialome recall is best in the herein described libraries. The reviewer has no doubt of the performance of presented screens, in fact they are illustrating the high degree of expertise in sgRNA screening accumulated in the Doench team. The authors attribute the effect mostly to new sgRNA design algorithms. However, screen performance also critically depends on multiple other parameters such as tracrRNA stem loop, Cas9 expression level, cell type and time of culture, equal representation of sgRNAs within the library, library representation in cells, MOI, PCR conditions for sgRNA retrieval from genomic DNA, NGS sequencing depth, and so forth. While the authors acknowledge such parameters in their discussion of results, they are not sufficiently incorporated in the side by side comparison of libraries as it is presented here.

To improve clarity and impact of this manuscript, the reviewer thus suggests:

a. A comparative table of all libraries discussed in this manuscript with regards to relevant parameters such as sgRNA representation bias, tracrRNA backbone, viral backbone, number of sgRNAs per gene, and so on. This includes the CRISPRa/i libraries discussed in the manuscript, where e.g. hCRISPRi-v2 uses another backbone than Dolcetto and it is likely that tracrRNA is relevant for CRISPRi performance.

b. To subtract screening parameters from sgRNA design, the authors should calculate sgRNA scores according to the new design rules onto other libraries such as Avana, GeCKO, Sabatini,

TKO, and Koike-Yusa, and others, to then demonstrate with analysis of AUC, that good sgRNAs within the older libraries indeed perform as the Brunello set. Importantly, it must be distinguished between datasets used to train the Brunello sgRNA design from independent datasets to prevent circular logic.

c. In the assumption that some sgRNAs within GeCKO, Avana, and Brunello are identical, the authors should use these sgRNAs as an indicator of screen performance independent on sgRNA design to control for above mentioned parameters.

d. While the CRISPRa/i libraries were subdivided into a set A and B in this manuscript, such division is lacking for Brunello. The authors should perform the same type of analysis on CRISPR Ko to show relevance of sgRNA design.

e. Alternatively or in addition, it would be interesting to clone a small library of essential genes based on different sgRNA designs and systematically compare them within a single experiment to avoid comparison between datasets variable in multiple additional parameters.

3) Lines 337-346: The comparison of number of "hits" between the SAM library and Calabrese is inadequate, as p-value cutoff might depend on e.g. NGS sequencing depth and other parameters. This is indicated by the fact, that at p-values $10e^{-3}$ to $10e^{-4}$, the number of called genes increases dramatically in Calabrese despite biological expectations. Interestingly, this presumably false positive calls do not occur at this interval for SAM. Please provide a better comparison.

4) The comparative discussion of CRISPRa in Figure 5 in general is based on few hits only and Pearson correlations between systems are very low as discussed by the authors. Therefore, a conclusive discussion of performance must include validation experiments of novel hits for genes solely identified in one CRISPRa screen but not another.

Point-by-point rebuttal

Reviewers' comments in blue

Authors' comments in black

We are glad that both reviewers believe this work will be of high interest and thank them for putting in the time to give a close reading of the manuscript and provide the constructive comments included below. Please note that all papers cited here in the rebuttal are also cited in the relevant section of the manuscript.

Reviewer #1 (Remarks to the Author):

Sanson, Hanna and colleagues present a comprehensive benchmarking of latest generation of human CRISPR libraries. They find that newest libraries deliver more accurate data with less sgRNAs. Metrics are developed by which to compare library performance. Other analyses demonstrate useful positional preferences for CRISPRi/a sgRNAs. A new tracr sequence is also developed and implemented for improved on-target performance.

Overall this is a key paper, holding a number of highly practical insights for the large and growing CRISPR screening field. Presentation and writing are top class.

Analyses are logical and intelligent.

Here I raise a number of issues which are relatively minor:

1. To what extent are differences observed between the authors' data and historical data (Fig1,3), due to (a) the potency of the libraries used, or (b) the expertise of the laboratory performing the experiments? (a) is assumed throughout the paper, but it would be worth at least raising the possibility of (b).

Reviewer 2 raises the same general point, and we agree that this is an important comparison to make. Originally, we addressed this questions specifically in reference only to CRISPRi technology (**Fig 3c**), where we show that screening results generated in another lab are improved if the analysis of sgRNAs is restricted to those most-preferred by our scoring scheme:

We found that the dAUC for this filtered subset of the hCRISPRi-v2 library was higher than the original hCRISPRi-v2 library, with the dAUC improving from 0.29 for all guides to 0.36 for this subset. We therefore conclude that the differences in performance between hCRISPRi-v2 and Dolcetto can mostly be attributed to improved CRISPRi sgRNA design rather than differences across cell lines or experimental execution.

In other words, when we applied our sgRNA selection scheme to outside data, we saw that it improved results, suggesting improvements in sgRNA design, at least for CRISPRi. We have now performed similar analyses for CRISPRko screens, from two perspectives.

First, we have now applied multiple sgRNA on-target scoring schemes to a common data set, the GeCKOv2 dataset from Project Achilles, which was screened across 33 cell lines. Importantly, this was the first library designed for CRISPR screens and the sgRNAs were selected without any knowledge of design rules (as they did not yet exist) and therefore

represents a neutral dataset on which to test various sgRNA scoring rules. Importantly for the analyses presented here, no data from these GeCKO screens, nor from any other negative selection screen, were used in creating Rule Set 2 or informed the selection of the sgRNAs in the Avana or Brunello libraries, so this analysis is truly a fair external dataset. However, the same is not true of other scoring schemes, some of which were based heavily on the GeCKO data itself or similar viability screens, so this metric may overestimate their generalizability. Nevertheless, we find that when GeCKOv2 screening data are filtered to include only the top 10% of sgRNAs as selected by each scoring scheme, Rule Set 2 showed the most improvement across 33 cell lines. We have included this analysis as **Fig. 1g**.

As a second line of evidence, we applied our sgRNA selection criteria to existing screening data conducted in other laboratories with other libraries. We show that filtering these datasets by using only the subset of sgRNAs that rank highly by our scoring criteria universally improves their performance (**Supplementary Fig. 1b**).

Between these two analyses, we believe that we have ruled out lab-specific executional expertise as a major factor in the differential performance. Please also see our response to Reviewer 2, point 1.

2. Tracr engineering: One is left slightly puzzled in the "Modifications to the tracrRNA" section. It is unclear firstly what is the motivation for changing the tracr in the first place. The take-home message seems to be that it yields a modest gain in performance, but with some downsides to off-targeting. I would not expect the authors to repeat any experiments, but perhaps to justify more clearly why this work was undertaken and fits in the present paper.

The reviewer succinctly summarizes our conclusion from these experiments. As to the rationale for this section, in the literature there are numerous claims that these modifications result in improved on-target activity¹⁻⁴. However, none of these manuscripts answered the question truly at-scale, nor did any examine whether or not off-target rates increase commensurately with on-target activity. And to be frank, whenever we present our work publically, this is one of the first questions from the audience, which tracrRNA we are using. We have thus modified the text to make the rationale for this section more obvious.

3. Use of FANTOM CAGE to identify TSS: This is one of the biggest doubts I have in the paper. - Firstly, the authors do not clearly explain exactly which CAGE dataset is used, even in the Methods. There are several different CAGE datasets, with different cutoffs and years of creation. This must be clearly specified.

To annotate the FANTOM5-defined TSS for each gene for use in designing CRISPRa and CRISPRi libraries, we used v1 of the reprocessed FANTOM5 CAGE data (accessed March 2016), in which the CAGE peaks from phase 1 and phase 2 were remapped to current genome assemblies (hg38 and mm10) using the liftOver tool as described in a publication from the FANTOM consortium⁵. Using the annotation file found in resource 1 (see below), we took the "CAGE Peak ID" for the highest ranked peak of every annotated gene in the FANTOM5 dataset, irrespective of the distance from the peak to Ensembl transcripts. We then searched for each CAGE Peak ID in the corresponding bed file (resource 2); the position marked as the "start of the representative TSS position" was considered the TSS for that gene. For the protein-coding genes that did not have TSS information in FANTOM5, we used the TSS of the principal transcript from Ensembl. If the TSS was not found in Ensembl, the TSS from NCBI was used.

We have updated the Methods section to include this information.

Resource 1:

http://fantom.gsc.riken.jp/5/datafiles/reprocessed/hg38_v1/extra/CAGE_peaks_annotation/hg38.cage_peak_phase1and2combined_liftover_ann.txt.gz

Resource 2:

http://fantom.gsc.riken.jp/5/datafiles/reprocessed/hg38_v1/extra/CAGE_peaks/hg38_liftover_CAGE_peaks_combined_phase1+2.bed.gz

Additionally, we emphasize that we are not claiming novelty in the use of FANTOM data for TSS annotation, as we are building directly off the results provided by Helin and colleagues⁶.

- Second: CAGE is an experimental method, performed in cells and tissues. Clearly therefore, differential TSS usage will be reflected in differential CAGE clusters between cell types. How does this influence the library designs performed for CRISPRi and CRISPRa libraries? How does this influence differences in apparent performance for the cell types shown in Fig3c,d? Specifically, it could be that HT29 TSS are simply worse reflected in the TSS dataset, compared to A375.

Helin and colleagues have shown that, at least in the K562 cell line, the majority of protein-coding genes do indeed use FANTOM-annotated p1 as the preferred transcript. To quote from that study:

We found that 87.4% of genes in K562 cells were predominantly expressed from p1 promoter, 8.3% from p2 promoters and 4.2% from p3 and higher (Figure 6). Thus, in situations where CAGE data is not available for a particular cell type, designing sgRNAs targeting the highest ranked promoters from the FANTOM5/CAGE atlas is expected to give the best chance of success.⁶

Because CAGE data are not available in FANTOM for cell lines screened with Dolcetto and Calabrese, we cannot compare how cell line differences in TSS annotations affect library performance. But this is a good general point to highlight as a limitation of these libraries and we have added it to the discussion. Of course, the more comprehensive one wants to be in the primary screening library (i.e. the more p2, p3, etc. promoters one chooses to target), the more sgRNAs one needs to add, greatly increasing the scale and thus cost of the screen.

- Third: It is also unclear in the Methods, how exactly FANTOM TSS were associated with gene TSS. In "CRISPRi and CRISPRa library design" it says that "we picked the TSS with the highest ranked peak..." - but in which length window? And what about the many genes with multiple TSS? I would invite the authors to much more clearly explain these crucial aspects of their method.

We did not perform any additional annotation of TSS sites beyond what was already annotated by FANTOM. Certainly this means that any inaccuracies or biases in the FANTOM dataset will also be reflected in our library designs. To quote from the original publication:

The first step, association with transcripts, was achieved by finding the TSS of transcripts within 500 bp flanking region of the CAGE peak (flanking regions are limited to 50 bp for 5'-end not derived from transcription initiation by RNA Polymerase II, such as small nucleolar RNA), choosing the nearest transcripts, and associating CAGE peaks to gene and protein models based on the nearest transcripts.⁵

If there are additional clarifications or caveats that this reviewer thinks would be useful for readers we would be happy to add them to the discussion.

- Fourth: This will influence the library design criteria. This should at least be much more clearly explained to the readers in the Discussion.

Indeed, one point we hope readers will take away from this manuscript is the importance of accurate TSS annotation for CRISPRa and CRISPRi experiments. To demonstrate this clearly, we specifically highlight an informative *failure* in our primary screen, missing BRAF as a resistance gene to MEK inhibition with CRISPRa. Only by expanding the window around the annotated TSS in the secondary screen were we able to capture sgRNAs that led to successful overexpression of BRAF (**Fig. 6g**). Although this is only an individual observation, we hope that it will serve as a forceful example of the importance of TSS annotation for success with this approach.

4. How TSS confidence influences knockdown: A question that is not addressed in this paper, is whether poor TSS annotation could explain poor knockdown in CRISPR screens. Of course it is up to the authors to decide whether to perform this or not. But I think the paper would improve, if some analysis were performed that broke down gene sets according to TSS confidence (ie how confidently the TSS is identified by CAGE), and the % effectiveness of its sgRNAs.

This is a great suggestion. As one metric of TSS confidence for each gene, we used the sum of CAGE reads per peak defined in the “score” column of resource 2 (see above) to calculate the fraction of all within-gene CAGE peak reads that belong to the selected TSS. The vast majority of genes had a relatively high TSS confidence by this measure. We then binned genes in both the essential and non-essential gene sets by TSS confidence and observed that genes with a higher TSS confidence showed greater depletion of essential genes, whereas the non-essential genes showed little change in activity (**Supplementary Fig. 4a, b**). Interestingly, only 58% of CAGE peaks are assigned to the p1 peak for BRAF, suggesting that low TSS confidence may explain the poor activity of the sgRNAs targeting this gene in the primary CRISPRa screen.

As a complementary approach, we also ranked the selected peaks for all genes in the Dolcetto library by the sum of their CAGE peak reads, and compared their percent ranks to the average log₂-fold change of essential and non-essential genes. We observe a similar trend by this metric for both the essential and non-essential genes (**Supplementary Fig. 4c, d**).

5. Figure 4A, the odd performance of Brunello in Non-Targeting genes: Could the authors offer any explanation for this strange result?

This is due to the so-called ‘cutting effect’ in which all cells that sense a dsDNA break temporarily slow down in their cell cycle (or, in extreme cases, undergo apoptosis). In the text, we write:

Conversely, non-targeting sgRNAs were among the least depleted, with an AUC = 0.16. That these sgRNAs are preferentially retained is expected based on the well-described cutting effect that manifests in CRISPRko screens, whereby dsDNA breaks caused by sgRNAs lead to a detectable effect on cell growth; in extreme cases, such as copy number amplified target sites or promiscuous sgRNAs, this effect is greatly magnified.

In the GeCKO library, a higher fraction of sgRNAs were poorly active, and thus also did not cut the genome, thereby ‘competing’ with non-targeting sgRNAs for enrichment at the top of the

screen. However, in the Brunello library, a much smaller fraction of sgRNAs fail to cut their target, and thus the enrichment at the top of the screen is more evident for the set of non-targeting sgRNAs in this dataset.

6. p12 / Figure 4 - discrepancy in CRISPRi / KO behaviour for histone genes: This is an interesting observation. Could any light be shed on it by studying other screen data, eg shRNA or siRNA datasets?

This is an excellent suggestion. We have added **Fig. 4g**, which examines the histone genes across the entire Cancer Dependency Map dataset, with hundreds of cell lines screened with both CRISPRko and RNAi technology. We continue to observe that only CRISPRko shows a viability effect with histone genes; we note that CERES scores are corrected for copy number amplification, further supporting our conclusion that this observation is not driven by a CRISPR cutting effect. However, essential genes are also differentially depleted between the technologies, likely reflecting the overall improvement in on-target activity from RNAi to CRISPRko. Therefore, although we cannot rule out the possibility that the differential depletion of histone genes simply reflects the overall ineffectiveness of RNAi reagents, this dataset further suggests that histone genes may more generally be poorly depleted through mechanisms that reduce, but do not eliminate, the mRNA. We readily admit that the mechanism behind this observation will require further study.

7. Figure 4B: better as a coloured density/intensity plot.

Yes, this is a good suggestion and we have modified this graph accordingly, as well as several others.

8. Figure 4C: axes are unclear, not sure what is NES.

Apologies for the jargon. NES stands for "Normalized Enrichment Score" and it is the preferred output of Gene Set Enrichment Analysis.

9. p.9 CAGE stands for "Cap Analysis of Gene Expression"

Corrected.

10. The authors might also briefly mention how results of this paper might impact other related fields, such as screens of long noncoding RNAs, and paired CRISPR deletion screens.

Yes, we have expanded the discussion to provide a wider use-case for these technologies.

Rory Johnson

Reviewer #2 (Remarks to the Author):

In the manuscript entitled "Up, down, and out: optimized libraries for CRISPRa, CRISPRi, and CRISPR-knockout genetic screens", Dr. Doench and Co-workers present an enormous amount of work including the generation of several genome wide sgRNA libraries against both human and mouse as well as 16 genome wide screens.

It is without a doubt that these libraries, already made accessible via addgene, will (continue to) receive great interest in the scientific community and become a widespread tool for genome

wide screens. For those reasons it is also of great interest to systematically compare library performance based on genome wide screens via a vis of technology (CRISPRa, CRISPRi, CRISPR ko), sgRNA design, and screen setup making this manuscript interesting and relevant in principle. The following section provides suggestions how to improve the comparison and clarity of the presented libraries.

We are glad that the reviewer appreciates the scale of data included here and believes this work will be of high interest.

1) The presented manuscript evaluates libraries based on performance of known positive (and negative) controls previously identified in similar sgRNA based screens. Like the control gene set itself, also the sgRNA design algorithm is based on previous screens and thus the used machine learning algorithms will bias towards known genes as well as specific guides depleting in CRISPR screens, thereby resulting in a somewhat circular argument and a self-fulfilling prophecy. It is thus pivotal to highlight this fact as a limitation of the method.

This is a very important point of clarification, and please also see our response to Reviewer 1, point 1, as it touches on some of these same issues.

It is certainly true that some other libraries relied on negative selection (viability) data to build their sgRNA selection strategy, but importantly, ours did not. First, our on-target rules for CRISPRko (Rule Set 2) derive from two data sets: Doench et al. 2014⁷, which examined a panel of cell surface receptors, and Doench et al. 2016⁸, which examined genes involved in small molecule resistance phenotypes (e.g. HPRT1 in cells treated with 6-thioguanine). None of the genes used to build Rule Set 2 are essential genes, nor were any results from screens conducted with the GeCKO, Avana, or any other libraries used to select sgRNAs for inclusion in the Brunello library.

Other libraries, however, did use viability data in the training of their selection criteria (e.g. Sabatini 2017) and/or selected sgRNAs based on previous past-performance (e.g. TKOv3 included guides that worked well in previous TKO libraries). The same is true for various scoring schemes that have been devised (e.g. Xu et al., 2015 uses viability data from Wang et al., 2014 screens to define the rules for sgRNA scores).

Nevertheless, despite these biases that might inflate the performance of other libraries and scoring schemes by the metrics used here, Brunello still outperforms them all.

Additionally, the reviewer is correct that the updated version⁹ of the essential gene set by Hart & Moffat had knowledge of initial CRISPR viability screens; however, the original version¹⁰ did not use any CRISPR screening data to define essential genes (the non-essential genes remain unchanged). We have added **Supplementary Fig. 1a** to show that the results shown in **Fig. 1c** are largely unchanged if one uses the original version of the essential gene list.

2) The manuscript compares many screens, old and new, performed by the authors as well as others, and derives values to show that e.g. the essentialome recall is best in the herein described libraries. The reviewer has no doubt of the performance of presented screens, in fact they are illustrating the high degree of expertise in sgRNA screening accumulated in the Doench team.

The authors attribute the effect mostly to new sgRNA design algorithms. However, screen performance also critically depends on multiple other parameters such as tracrRNA stem loop,

Cas9 expression level, cell type and time of culture, equal representation of sgRNAs within the library, library representation in cells, MOI, PCR conditions for sgRNA retrieval from genomic DNA, NGS sequencing depth, and so forth. While the authors acknowledge such parameters in their discussion of results, they are not sufficiently incorporated in the side by side comparison of libraries as it is presented here.

To improve clarity and impact of this manuscript, the reviewer thus suggests:

a. A comparative table of all libraries discussed in this manuscript with regards to relevant parameters such as sgRNA representation bias, tracrRNA backbone, viral backbone, number of sgRNAs per gene, and so on. This includes the CRISPRa/i libraries discussed in the manuscript, where e.g. hCRISPRi-v2 uses another backbone than Dolcetto and it is likely that tracrRNA is relevant for CRIPRi performance.

We have compiled this information to the best of our ability (some experimental details are difficult to discern in published manuscripts). These are provided as **Supplementary Tables 1, 3, and 5**. Given how many variables there are, and how few studies there are relative to the number of variables, we do not think it possible to unambiguously prove a causal relationship between these experimental factors and ultimate performance purely on the basis of a meta-analysis. However, we hope that some readers will find the compilation of these factors useful.

Additionally, please note that, for CRISPRi, we use a similar modified tracrRNA to that used in hCRISPRi-v2. From the text:

Here, we opted to use tracr-v2 for CRISPRi both because off-target effects are potentially less of a concern due to the limited window of activity and because previous CRISPRi studies have used a modified tracrRNA structure.

b. To subtract screening parameters from sgRNA design, the authors should calculate sgRNA scores according to the new design rules onto other libraries such as Avana, GeCKO, Sabatini, TKO, and Koike-Yusa, and others, to then demonstrate with analysis of AUC, that good sgRNAs within the older libraries indeed perform as the Brunello set. Importantly, it must be distinguished between datasets used to train the Brunello sgRNA design from independent datasets to prevent circular logic.

Yes, this is an excellent suggestion to distinguish technical expertise in this group from actual improvements to library design, and we have added these analyses as **Fig. 1g** and **Supplementary Fig. 1b**. Additionally, we emphasize that Brunello sgRNAs designs are in no way based on previous viability screens, so there is no cause for concern of circular logic on this specific point. Please also see our response to Reviewer 1, point 1.

c. In the assumption that some sgRNAs within GeCKO, Avana, and Brunello are identical, the authors should use these sgRNAs as an indicator of screen performance independent on sgRNA design to control for above mentioned parameters.

We compared the performance of the several thousand sgRNAs in common between the GeCKO and Brunello libraries, and the Pearson correlation is 0.58. As we are not quite sure how to contextualize that number for readers, we believe that the cross-library and cross-scoring-scheme analyses we have included are a more forceful and intuitive approach to control for difference in screening parameters across laboratories.

d. While the CRISPRa/i libraries were subdivided into a set A and B in this manuscript, such division is lacking for Brunello. The authors should perform the same type of analysis on CRISPR Ko to show relevance of sgRNA design.

We have performed this analysis, and the results are provided here, grouping the top two sgRNAs per gene into Set A and the next two into Set B. The plot is shown to the same scale as **Fig. 1c**:

It is important to note that the universe of sgRNAs under consideration for a CRISPRko library is much larger than that for CRISPRa and CRISPRi, as much of the coding sequence can be a productive target, while the latter designs are much more constrained by the location of the sgRNAs relative to the TSS. Thus, one would not necessarily expect to have the same discrimination at the top of the list of CRISPRko sgRNAs (i.e. guides 1 - 4 out of hundreds of possibilities) whereas for CRISPRa/i, one is selecting from only a couple dozen possibilities. We have currently chosen not to include this specific analysis in the text, as we believe that the existing analyses thoroughly bolster our conclusions about sgRNA selection, but would be happy to add it if the reviewer or editor feels strongly.

e. Alternatively or in addition, it would be interesting to clone a small library of essential genes based on different sgRNA designs and systematically compare them within a single experiment to avoid comparison between datasets variable in multiple additional parameters.

We have thought about this experiment, but are concerned that it could be confounded by the point raised previously, namely that if a scoring scheme was already based on negative selection screens with essential genes, then its performance in this comparison would be biased.

3) Lines 337-346: The comparison of number of “hits” between the SAM library and Calabrese is inadequate, as p-value cutoff might depend on e.g. NGS sequencing depth and other parameters. This is indicated by the fact, that at p-values $10e^{-3}$ to $10e^{-4}$, the number of called genes increases dramatically in Calabrese despite biological expectations. Interestingly, this presumably false positive calls do not occur at this interval for SAM. Please provide a better comparison.

The reviewer is correct that the comparison of these screens could be due to quality of the screens rather than the library itself. We do not quite understand what the reviewer means by the “biological expectations” of the distribution of p-values, as generally we see that different screens can have very different distributions when plotted this way (e.g. some screens have very few genes score quite strongly, some have larger numbers of genes with intermediate phenotypes, etc.) -- some of this will be intrinsic to the phenotype (how many genes are actually involved, and the magnitude of that signal) whereas others will be due to the execution of the screen (poorer technical execution will lead to fewer strong hits). Regardless, the key point raised by the reviewer is whether these genes are, in fact, false positive calls.

To better examine this, we conducted a secondary follow-up screen with more sgRNAs per gene (**Fig. 5e**). These screens showed good reproducibility, likely due to the smaller scale and subsequent sgRNA coverage per cell; we achieved a gene-level Pearson correlation of 0.84 when comparing dCas9 to dCas9-VP64 (**Fig. 5f**). Examining all the genes that scored with a p-value of 10^{-3} or better in the primary screen, we found that 73% of the genes (41 of 56)

nominated by the Calabrese library validated in the secondary screen, at a stringent FDR cut-off of < 5% (**Fig. 5g**). In this same secondary screen, we validated 10 of the 17 genes (59%) nominated by the SAM screen. This shows that many of the genes nominated in this “intermediate” p-value range of 10^{-3} to 10^{-4} were in fact true positives in the Calabrese screen.

We note that this analysis does not conclusively rule out that the screens with the SAM library were conducted to a lower technical standard than those presented here, but it does at least suggest that the false positive rate of the primary screen conducted with Calabrese is reasonably low.

4) The comparative discussion of CRISPRa in Figure 5 in general is based on few hits only and Pearson correlations between systems are very low as discussed by the authors. Therefore, a conclusive discussion of performance must include validation experiments of novel hits for genes solely identified in one CRISPRa screen but not another.

This comment builds off the previous one. With CRISPRa, unlike CRISPRko and CRISPRi with the set of essential and non-essential genes, there is no ground truth, and thus comparisons are necessarily made based on modest numbers of hit genes. Indeed, the comparisons to the ORF library screened by Elledge and colleagues (**Fig. 6h**) trends in the right direction, but is hardly conclusive.

Thus, we have attempted to provide more confidence for the genes that score in the selumetinib resistance screen. A secondary screen of genes scoring in either the primary ORF or CRISPRa screen with trametinib or selumetinib, respectively, revealed that the validation rate for genes tracked largely with the p-values in the primary screen (**Fig. 6e**). This secondary screen also made clear that differences between ORF and CRISPRa technology were not due simply to technical noise in the CRISPRa screen, as the secondary screens were quite reproducible across small molecules (**Fig. 6d**), but rather likely stem from true differences in the nature of the perturbations. To give one example, the sgRNAs originally designed against BRAF failed to score in the primary screen with Calabrese (**Fig. 6g**) and only upon selecting more sgRNAs across the promoter, in the secondary screen, were we able to validate this gene as a hit.

In sum, although we cannot claim to know how close to the finish line we are with CRISPRa technology, we hope that these results are convincing that the Calabrese library is currently the best-in-class for this technology.

REFERENCES

1. Cross, B. C. S. *et al.* Increasing the performance of pooled CRISPR-Cas9 drop-out screening. *Sci. Rep.* **6**, 31782 (2016).
2. Chen, B. *et al.* Dynamic imaging of genomic loci in living human cells by an optimized CRISPR/Cas system. *Cell* **155**, 1479–1491 (2013).
3. Dang, Y. *et al.* Optimizing sgRNA structure to improve CRISPR-Cas9 knockout efficiency. *Genome Biol.* **16**, 280 (2015).
4. Tzelepis, K. *et al.* A CRISPR Dropout Screen Identifies Genetic Vulnerabilities and Therapeutic Targets in Acute Myeloid Leukemia. *Cell Rep.* **17**, 1193–1205 (2016).
5. Abugessaisa, I. *et al.* FANTOM5 CAGE profiles of human and mouse reprocessed for GRCh38 and GRCm38 genome assemblies. *Sci Data* **4**, 170107 (2017).
6. Radzisheuskaya, A., Shlyueva, D., Müller, I. & Helin, K. Optimizing sgRNA position markedly improves the efficiency of CRISPR/dCas9-mediated transcriptional repression. *Nucleic Acids Res.* **44**, e141 (2016).
7. Doench, J. G. *et al.* Rational design of highly active sgRNAs for CRISPR-Cas9-mediated gene inactivation. *Nat. Biotechnol.* **32**, 1262–1267 (2014).
8. Doench, J. G. *et al.* Optimized sgRNA design to maximize activity and minimize off-target effects of CRISPR-Cas9. *Nat. Biotechnol.* **34**, 184–191 (2016).
9. Hart, T. *et al.* High-Resolution CRISPR Screens Reveal Fitness Genes and Genotype-Specific Cancer Liabilities. *Cell* **163**, 1515–1526 (2015).
10. Hart, T., Brown, K. R., Sircoulomb, F., Rottapel, R. & Moffat, J. Measuring error rates in genomic perturbation screens: gold standards for human functional genomics. *Mol. Syst. Biol.* **10**, 733 (2014).

Reviewers' Comments:

Reviewer #1:

Remarks to the Author:

I thank the Authors for responding to my comments.

Reviewer #2:

Remarks to the Author:

We thank the authors for careful consideration and implementation of all reviewers comments, in particular figure 1g is a strong addition. Together with new figure 1h incorporating also very recent publications on editing outcomes, these modifications clearly further improved the manuscript. We also thank the authors for including the overview table S1, which is helpful to interpret across previously published screens.

In summary, this manuscript reports an amazing quantity of work, provides a powerful resource and analysis to the community, will be highly cited, and I thus very strongly support publication.

We congratulate Kendall Sanson, Ruth Hanna, and all other co-authors to this work and hope it will receive the attention that it deserves.

Point-by-point rebuttal

Reviewers' comments in blue

Authors' comments in black

Reviewer #1 (Remarks to the Author):

I thank the Authors for responding to my comments.

We thank the reviewer for taking the time to carefully review our resubmission and respond to the new analyses and data we provided and for his helpful comments throughout the review process.

Reviewer #2 (Remarks to the Author):

We thank the authors for careful consideration and implementation of all reviewers comments, in particular figure 1g is a strong addition. Together with new figure 1h incorporating also very recent publications on editing outcomes, these modifications clearly further improved the manuscript. We also thank the authors for including the overview table S1, which is helpful to interpret across previously published screens.

In summary, this manuscript reports an amazing quantity of work, provides a powerful resource and analysis to the community, will be highly cited, and I thus very strongly support publication.

We congratulate Kendall Sanson, Ruth Hanna, and all other co-authors to this work and hope it will receive the attention that it deserves.

Ulrich Elling

We thank the reviewer for the constructive feedback and suggestions to better improve the manuscript throughout the process and accept your congratulations. Thank you for taking the time to review our resubmission.